# Altered grid-like coding in early blind people

Federica Sigismondi [1,3] ✉, Yangwen Xu [1,2,3], Mattia Silvestri[1] & Roberto Bottini [1] ✉

Cognitive maps in the hippocampal-entorhinal system are central for the representation of both spatial and non-spatial relationships. Although this system, especially in humans, heavily relies on vision, the role of visual experience in shaping the development of cognitive maps remains largely unknown. Here, we test sighted and early blind individuals in both imagined navigation in fMRI and real-world navigation. During imagined navigation, the Human Navigation Network, constituted by frontal, medial temporal, and parietal cortices, is reliably activated in both groups, showing resilience to visual deprivation. However, neural geometry analyses highlight crucial differences between groups. A 60° rotational symmetry, characteristic of a hexagonal grid-like coding, emerges in the entorhinal cortex of sighted but not blind people, who instead show a 90° (4-fold) symmetry, indicative of a square grid. Moreover, higher parietal cortex activity during navigation in blind people correlates with the magnitude of 4-fold symmetry. In sum, early blindness can alter the geometry of entorhinal cognitive maps, possibly as a consequence of higher reliance on parietal egocentric coding during navigation.

Humans heavily rely on vision to navigate the environment and acquire spatial information[1-4]. However, the role of visual experience in the development and functioning of the brain system dedicated to spatial navigation remains largely unknown. In the brain, the human navigation network (HNN) comprises medial temporal, parietal, and occipital regions that support spatial memory and navigation across complementary allocentric and egocentric reference frames[5-11]. Significant breakthroughs in rodent electrophysiology attribute a crucial role in spatial navigation to the hippocampal-entorhinal system through the construction of allocentric cognitive maps[12] based on the activity of highly specialized neurons, such as place cells[13], head direction cells[14], and grid cells[15]. Whereas place cells and head direction cells encode single locations and heading directions, respectively, grid cells in the medial entorhinal cortex (EC) present multiple firing fields and keep track of agent movements within the environment, tiling the navigable surface in a regular hexagonal grid.

Compared to other modalities, vision provides highly precise and global information regarding the location of both proximal and distal landmarks, which are important for the creation of allocentric maps[16-18]. Notably, visual landmarks displaced in the environment as

experimental manipulation cause a shift in the firing field orientation of grid, place, and head direction cells toward the position of the cues, suggesting visual anchoring of cognitive maps[15,19-22]. Consistently, when rodents need to navigate in the dark, after familiarizing themselves with an environment in plain light, both grid and head direction firing fields can be disrupted and navigation impaired[23,24]. However, experiments with congenitally blind rodents have shown that place cell firing fields develop normally in the absence of vision, although with reduced firing rates[25], and head direction cells maintain their directional selectivity but with lower precision[24]. Nothing is known about the resilience of place and head direction coding in blind humans, and the development of grid cells in the absence of functional vision remains untested across species. Notably, in the lack of vision, rodents can maintain stable allocentric maps of the environment by relying on olfactory cues[24], a sense that, arguably, is not sufficiently developed in humans to serve the same purpose. How does early or congenital visual deprivation affect spatial navigation and its neural underpinnings in humans?

The ability to create mental maps of the environment is maintained in blind people[26,27], though early lack of vision can impair

[1]Center for Mind/Brain Sciences, University of Trento, 38122 Trento, Italy. [2]Max Planck Institute for Human Cognitive and Brain Sciences, D-04303 Leipzig, Germany. [3]These authors contributed equally: Federica Sigismondi, Yangwen Xu. ✉e-mail: federica.sigismondi@unitn.it; roberto.bottini@unitn.it

allocentric spatial coding[28], leading to less reliable distance estimation[29] and higher reliance on egocentric coding[27,30]. A few neuroimaging studies have investigated the influence of early visual deprivation on the neural correlates underlying spatial navigation in early blind individuals using various tasks and reporting inconsistent results[31–35] (see "Discussion" section). Crucially, none of these experiments has investigated the impact of blindness on the hippocampal-entorhinal spatial codes.

Here, in an fMRI experiment, we asked blindfolded sighted and early blind participants to imagine navigating inside a clock-like space, walking from one number to another in a straight line, to investigate whether early lack of vision influenced the emergence of grid-like representations in early blind individuals' entorhinal cortex. The clock space was sampled at a granularity of 15° to provide a sufficient angle resolution to detect the hexadirectional signal (6-fold symmetry), which some studies[36–40] have indicated as a proxy for the activity of grid cells in fMRI both during visual[36–39] and imagined navigation[40,41]. Participants' active navigation during the task was spurred by the introduction of a control question, solvable only by performing spatial inferences on the relationships between the positions of the different numbers inside the clock. Importantly, we've benefitted of the high familiarity of both groups with the clock environment to control for a possible imbalance of task-familiarity. We also ran a shorter, modified version of the experiment, in which clock navigation was compared to an arithmetic task (Nav-Math Experiment) to test possible differences

between sighted and early blind individuals in the activation of navigation-specific regions. Finally, we tested how brain activations relate to real-world spatial navigation in both groups.

## Results

### Blind and sighted individuals successfully navigated the clock space

The imagined navigation task was similar across the two experiments (Clock-Navigation and Nav-Math), (See "Methods" section and Fig. 1A, B), and was developed by adapting imagined navigation tasks from previous studies[40,41]. Blindfolded participants received auditory instructions for their starting and target locations (numbers) inside the clock. They were then told to imagine walking directly from one point to the other. After a jittered imagination period (4–6 s), a third number was announced, and participants had to decide whether the number was located to their left or right. The arithmetic task (performed only in the Nav-Math experiment) was structurally similar to the navigation one. Participants heard three numbers, on which they had to perform simple operations (e.g., 8 + 2 − 4) and then compared the results to a target number presented at the beginning of each block (Fig. 1B). Nineteen early blind participants (who lost completely their sight at birth or before 5 y.o., and report no visual memory; Table S1) and 19 matched sighted controls underwent an fMRI session, preceded by behavioral training in both tasks (math and navigation) and experiments (Nav-Math and Clock-Navigation, see "Methods" section). No

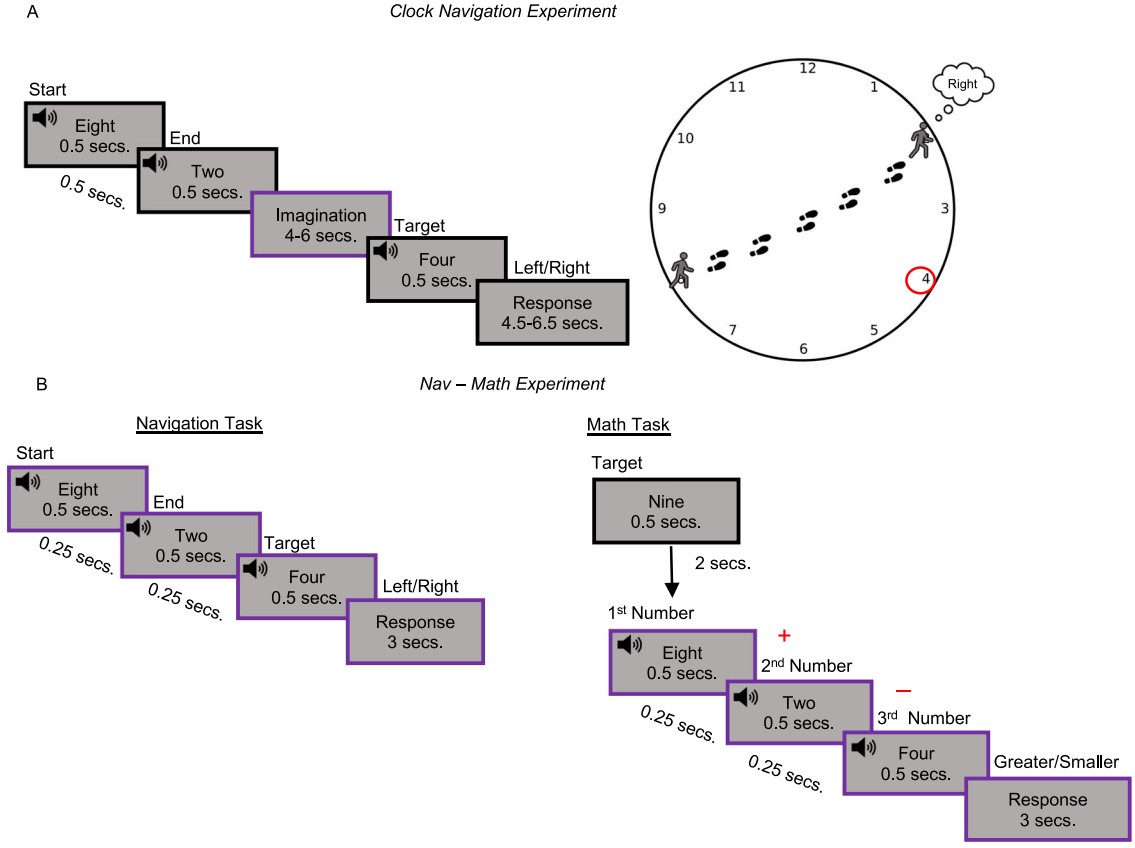

**Fig. 1 | Clock-Navigation and Nav-Math experimental timelines. A** Timeline of the Clock-Navigation experiment: participants were asked to imagine navigating from one number (Starting point) to another (Ending point) in the clock space, according to auditory instructions. Successful trials were denoted as those in which participants correctly indicated the position (left or right) of a third number (Target number) in the space, compared to their current trajectory, as shown in the graphical representation on the right. The purple box indicates the event of interest for the Quadrature Filter analysis. **B** Timeline of the Nav-Math experiment: the navigation task structure (left) was similar to that of the Clock Navigation experiment,

with timing being the only difference. During the Math task (right), participants heard a number at the beginning of each math block (Target number) and then three more numbers. Participants were asked to sum the first two numbers, subtract the third number from that result, and then to compare the results with the target number. Importantly, the navigation and mathematic tasks used the exact same numerical stimuli (see Fig. 1B) with different instructions. Purple boxes indicate the periods used for the univariate analysis (participant reaction times were used to determine the duration of the response period).

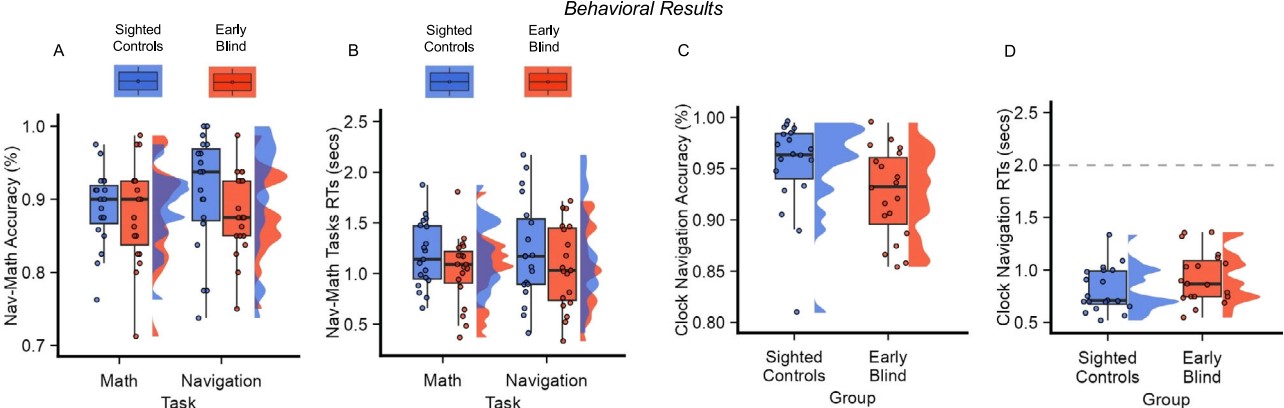

**Fig. 2 | Blind and sighted individuals successfully navigated the clock space. A**, **C** Accuracy of the two groups in the Nav-Math (**A**) and the Clock Navigation (**C**) experiments. Successful trials were counted as those in which participants correctly responded to the control question (Left/Right or Greater/Smaller). No difference in performance was detected between the two groups in the Clock-Navigation experiment (Two-sample *t*-test; $n = 38$, $p = 0.052$, two-tailed), nor between groups and tasks in the Nav-Math experiment (Repeated-measure ANOVA; $n = 38$, $p = 0.24$, two-tailed). **B**, **D** Reaction Times (RTs) of the two groups (sighted controls in blue and early blind individuals in red). In the Nav-Math experiment, participants were asked to give a response as quickly as possible. No differences were found between

groups and tasks (Linear Mixed-Effect Model; $n = 38$, $p = 0.61$, two-tailed). In the Clock-Navigation experiment, all participants answered before the time limit (gray dotted line), with no difference between groups (Linear Mixed-Effect Model; $n = 38$, $p = 0.11$, two-tailed). Boxes indicate the interquartile Range (IQR, data points included between the first quartile, 25th percentile, and the third quartile, 75th percentile), the horizontal black line indicates the median (50th percentile), and the whiskers the distance between the first and third quartile to highest and the lowest value in the sample. The distribution of the data is represented by a density curve beside each box. Significance levels are defined as follow: *$p < 0.05$; **$p < 0.01$; ***$p < 0.001$. Source data are provided as a Source Data file.

difference in accuracy was detected between groups and tasks in the Nav-Math experiment (Repeated-measure ANOVA; main effect of group: $F(1,36) = 0.86$, $p = 0.36$, $\eta^2 = 0.018$, 95% CI = [−0.04, 0.03]; main effect of task: $F(1,36) = 0.85$, $p = 0.36$, $\eta^2 = 0.005$, 95% CI = [−0.006, 0.04]; Group × Task interaction: $F(1,36) = 1.42$, $p = 0.24$, $\eta^2 = 0.008$, 95% CI = [−0.06, 0.01], two-tailed; Fig. 2A). Reaction times (RTs) followed a similar pattern with no differences detected in RTs between groups and tasks in the Nav-Math experiment (main effect of group $\chi^2(1) = 1.53$, $p = 0.21$, 95% CI [−0.39, 0.08]; main effect of task: $\chi^2(1) = 3.59$, $p = 0.06$, 95% CI [−0.02, 0.07]; Group × Task interaction: $\chi^2(1) = 0.24$, $p = 0.62$, 95% CI [−0.04, 0.08], two-tailed, Fig. 2B). Collectively, these results suggest that both sighted and blind participants were able to perform the tasks and to perform comparably.

Similarly, in the Clock-Navigation experiment, we did not detect differences in accuracy (two-sample *t*-test; $t(36) = 2.0$, $p = 0.052$, Cohen's $d = 0.62$, 95% CI = [−0.0018, 0.057], two-tailed, Fig. 2C) and RTs (Linear Mixed-Effect Model; $\chi^2(1) = 2.38$, $p = 0.12$, 95% CI = [−0.03, 0.26], two-tailed, Fig. 2D) between groups. However, outlier analysis revealed the presence of one outlier in accuracy in the sighted control group. Thus, we've performed the analyses excluding the outlier participants. Results on the RTs remain invariant with no differences between groups (Linear Mixed-Effect Model; 18 sighted controls & 19 early blind individuals, $\chi^2(1) = 2.51$, $p = 0.13$, 95% CI = [−0.02, 0.28] two-tailed). On the other hand, analysis on accuracy detected a significant difference between the two groups (18 sighted controls & 19 early blind individuals, $t(35) = 2.87$, $p = 0.007$, Cohen's $d = 0.94$, 95% CI = [0.01, 0.06], two-tailed). Despite this slight difference in navigation accuracy, it is important to consider that the average accuracy was very high in both groups (19 sighted controls, accuracy % = 95; 19 early blind individuals, accuracy % = 92), therefore we could conclude that both sighted and blind participants were able to successfully navigate the clock environment during the Clock-Navigation experiment.

**The human navigation network is resilient to early visual deprivation**

We analyzed fMRI data from the Nav-Math experiment to investigate whether early blind individuals and sighted controls rely on the same brain network typically involved in navigation (HNN). A brain mask, downloaded from Neurosynth, comprising brain regions in the HNN

(Fig. S1, see "Methods" section) was used to detect regions with a greater level of activation during navigation compared to mathematics. The mask included (bilaterally) the middle frontal gyrus (MFG), superior parietal lobe/precuneus (SPL), retrosplenial cortex (RSC), occipital place area (OPA), parahippocampal place area (PPA) and hippocampus (HC).

The univariate contrast (i.e., Navigation > Math) revealed significant activity in all the ROIs, except for the hippocampus, in both sighted and blind people (Fig. 3A, B and Table S1, all results thresholded at $p_{FDR} < 0.05$). No significant cluster of activation emerged when we contrasted the two groups ([Navigation > Math] × [early blind vs. sighted controls]). Highly similar results emerged from unmasked whole-brain analysis (Fig. S2). Given that previous studies have found significant differences between sighted controls and early blind people during navigation tasks using univariate contrasts[31,32], we performed a whole-brain analysis to investigate whether some differences emerged across groups in regions that were not included in our HNN mask. This analysis did not reveal any significant difference between groups when correcting for multiple comparisons ($p_{FDR} < 0.05$). Since previous studies reported that blind individuals tend to rely more on egocentric coding of space compared to sighted individuals[27,28,30], we conducted a Small Volume Correction (SVC) analysis within the inferior parietal cortex (IPC), a region that has been implicated in the egocentric coding of spatial and non-spatial information[42,43]. We used a spherical ROI (Radius = 10 mm) centered on Montreal Neurological Institute (MNI) peak coordinates (36/−68/44) obtained from an independent study investigating egocentric spatial representations during imagined navigation in a circular environment, structurally similar to our clock[44]. The analysis revealed a significant difference during imagined navigation between sighted controls and early blind individuals within the inferior parietal cortex ([Navigation > Math] × [early blind > sighted controls]; SVC: Voxel-level $p < 0.001$, Cluster-level $p_{FWE} < 0.05$; see "Methods" section and Fig. S3). Supporting the SVC analysis, whole-brain results using a lenient threshold ($p < 0.005$ uncorrected) revealed the emergence of a cluster of activation in the right inferior parietal cortex (IPC, 39/−61/53, Fig. 3C), overlapping with our ROI. We then used this inferior parietal cluster as an independently localized ROI for analysis in the Clock-Navigation experiment to test for possible parietal-based compensation during navigation in the early blind group.

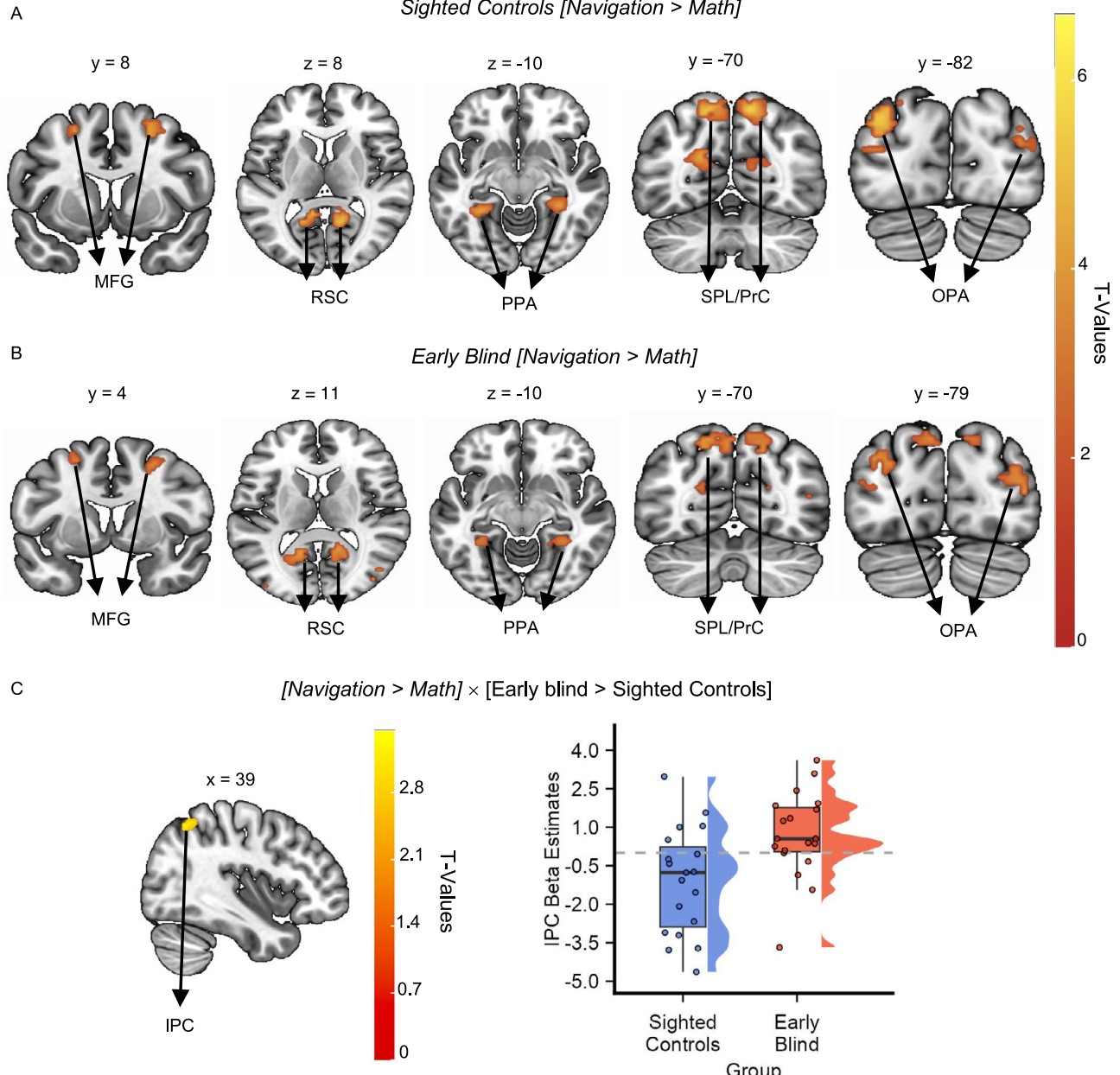

**Fig. 3 | The human navigation network is resilient to early visual deprivation. A, B** Results of the univariate analysis [Navigation > Math] performed within each group on the Nav-Math experiment data: the results showed activation in both sighted controls (**A**, *n* = 19) and early blind (**B**, *n* = 19) participants in the HNN mask (the activations were thresholded at $p_{FDR}$ < 0.05 and overlapped with the MNI-152 T1 template). Middle frontal gyrus (MFG), superior parietal lobe/precuneus (SPL), retrosplenial cortex (RSC), occipital place area (OPA), parahippocampal place area (PPA). **C** Whole-brain (left), group-level univariate analysis ([Navigation > Math] × [early blind > sighted controls]) on unmasked data revealed higher activation in the inferior parietal cortex in early blind individuals compared to sighted control individuals when observed using a lenient threshold (IPC, 39/−61/53, Two-sample

*t*-test, *n* = 38, *p* < 0.005 uncorrected, activation overlapped with the MNI-152 T1 template). Right: beta estimates extracted within the IPC cluster (Navigation > Math) for each group (for display only) show that the parietal cortex was more involved in the navigation task in the early blind group. Source data are provided as a Source Data file. The boxes indicate the IQR (data points included between the first quartile, 25th percentile, and the third quartile, 75th percentile), the horizontal black line the median (50th percentile), and the whiskers the distance between the first and third quartile to the highest and the lowest value in the sample. The distribution of the data is represented by a density curve beside each box.

## Six-fold grid-like coding did not emerge in early blind individuals

Next, we investigated the stability of grid-like coding in the EC of both groups by analyzing the data from the Clock-Navigation Experiment. In fMRI, grid-like coding can be detected as a 60° sinusoidal modulation of the Blood-Oxygen-Level-Dependent (BOLD) signal (6-fold symmetry, Fig. 4A, B), elicited by spatial trajectories aligned with the main axis of the hexagonal grid[36]. We thus applied a

four-way cross-validation procedure and implemented quadrature filter analysis in which three partitions (six runs) of the data were used to estimate the subject-specific grid orientation (phase, φ), and the remaining partition (two runs) was used to test the strength of the 60° sinusoidal modulation of the BOLD signal in EC after realigning the trajectories' angles to each participant's specific grid orientation[36,37] (see "Methods" section). Considering the lack of an a priori hypothesis on grid-coding lateralization, we first combined all

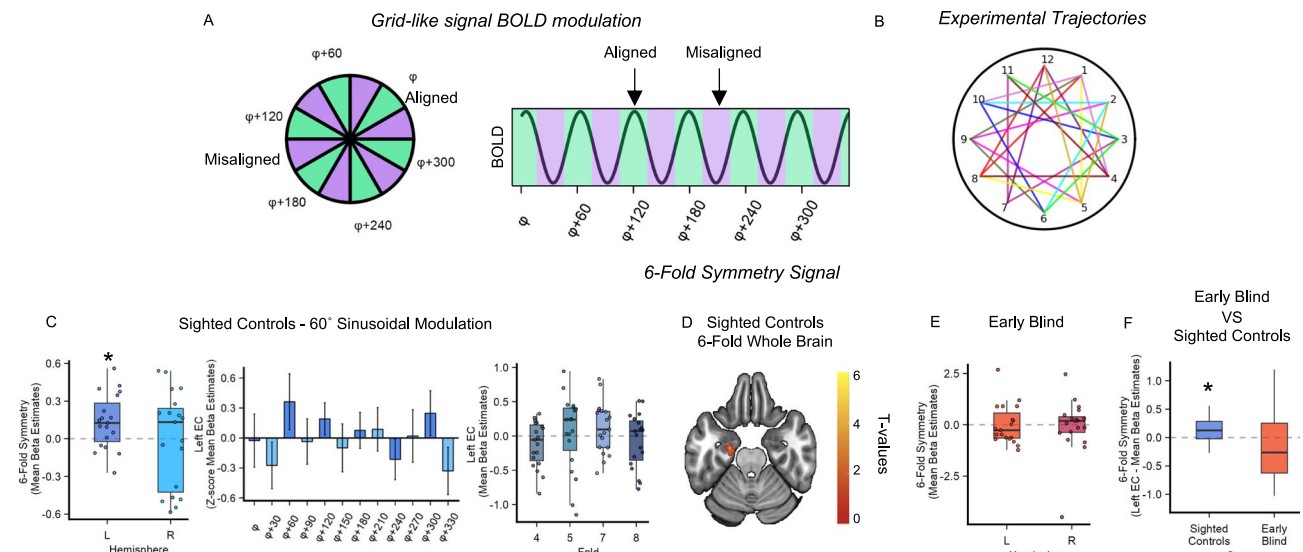

**Fig. 4 | Six-fold grid-like coding does not emerge in early blind individuals' EC.**
**A** Left: grid cells fire more when the navigator moves in a direction aligned (green) with the preferred grid orientation (φ) and its 60° multiples, compared to misaligned directions (purple). Right: grid-like coding can be detected using fMRI as a 60° sinusoidal modulation of the BOLD signal (6-fold symmetry), as the conjunctive activation of grid cells produced a higher BOLD modulation for movements aligned rather than misaligned with the main grid orientation. **B** Participants performed 24 trajectories in each run, sampling the space with a granularity of 15°. Different colors represent different experimental trajectories, the same colors have been used to identify experimental trajectories with a common starting/ending point. **C** Quadrature filter analysis revealed the presence of a significant 6-fold symmetry in the sighted controls' left EC (left, one-sample *t*-test; *n* = 19, *p* = 0.007, $\alpha_{Bonferroni}$ = 0.016, one-tailed), noticeable also as a 60° sinusoidal modulation of the EC BOLD signal (Data are presented as mean value ± SEM, for display only). No significant BOLD modulations were found in the tested alternative periodicities

(right panel). Source data are provided as a Source Data file. **D** Whole-brain, group-level analyses revealed a cluster of activation in the left EC of sighted participants for the periodicity of interest (60°; the activation overlaps with the MNI-152 T1 template at an uncorrected threshold of *p* < 0.01 voxel-level, one-sample *t*-test), for display only. **E** No 6-fold symmetry modulation was detected in the early blind individuals' EC (Wilcoxon signed-rank test; *n* = 19, Left EC: *p* = 0.37, and Right EC: *p* = 0.37, all results one-tailed, α = 0.05). Source data are provided as a Source Data file. **F** No differences between groups were observed (Wilcoxon test; *n* = 38, *p* = 0.14, *r* = 0.24, α = 0.05, two-tailed). Source data are provided as a Source Data file. In box plots, the boxes indicate the IQR (data points included between the first quartile, 25th percentile, and the third quartile, 75th percentile), the horizontal black line the median (50th percentile), and the whiskers the distance between the first and third quartile to highest and the lowest value in the sample. Significance levels are defined as follows: *\*p* < 0.05; *\*\*p* < 0.01; *\*\*\*p* < 0.001.

sighted controls' bilateral ECs into a single ROI, thus estimating a common grid orientation for the left and the right EC[37]. No significant 60° sinusoidal modulation (6-fold symmetry) of the BOLD signal was found in the sighted bilateral EC (One-sample *t*-test; *t*(18) = 1.07, *p* = 0.30, Cohen's *d* = 0.24, 95% CI = [−0.06, 0.2], α = 0.05, two-tailed). We then analyzed the two hemispheres separately and found a significant signature of grid-like coding in the sighted controls' left EC (Bonferroni-corrected for multiple comparisons across left, right, and bilateral hemispheres; one-sample *t*-test; *t*(18) = 2.71, *p* = 0.007, Cohen's *d* = 0.62, 95% CI = [0.05, Inf], one-tailed, $\alpha_{Bonferroni}$ = 0.016; Fig. 4C, D) but not in the right EC (Bonferroni-corrected for multiple comparisons across left, right and bilateral hemispheres; one-sample *t*-test; *t*(18) = −0.04, p = 0.5, Cohen's *d* = 0.009, 95% CI = [−0.15, Inf], one-tailed, $\alpha_{Bonferroni}$ = 0.016; Fig. 4C, D). Interestingly, the left lateralization of the grid-like signal was also reported in the two previous studies on imagined navigation[40,41]. Equal analyses performed on alternative models in the left EC (4-,5-,7-, and 8-fold symmetry) confirmed the specificity of our 6-fold symmetry results (Bonferroni-corrected across left, right and bilateral hemispheres; one-sample t-test; 4-fold: *t*(18) = −1.58, *p* = 0.93, Cohen's *d* = 0.36, 95% CI = [−0.25, Inf]; 5-fold: *t*(18) = 0.35, *p* = 0.37, Cohen's *d* = 0.08, 95% CI = [−0.18, Inf]; 7-fold: *t*(18) = 1.38, *p* = 0.09, Cohen's d = 0.31, 95% CI = [−0.02, Inf], and 8-fold: *t*(18) = −0.73, *p* = 0.76 Cohen's *d* = 0.17, 95% CI = [−0.22, Inf],all results one-tailed, $\alpha_{Bonferroni}$ = 0.016; Fig. 4C). Moreover, we did not detect any significant modulation of the BOLD signal when alternative models were tested in the right EC (Bonferroni-corrected across left, right and bilateral hemispheres; 4-fold: t(18) = −0.86, *p* = 0.8, Cohen's *d* = 0.20, 95% CI = [−0.2, Inf]; 5-fold: t(18) = 1.23, *p* = 0.11, Cohen's *d* = 0.28, 95% CI = [−0.06, Inf]; 7-fold:

*t*(18) = 1.56, *p* = 0.07, Cohen's *d* = 0.35, 95% CI = [−0.05, 0.36], and 8-fold: *t*(18) = −0.54, *p* = 0.7, Cohen's *d* = 0.12, 95%CI = [−0.23, Inf],all results one-tailed, $\alpha_{Bonferroni}$ = 0.016). The same analysis was conducted on the early blind population. Accounting for the non-normal distribution of 6-fold symmetry beta estimates in this group (Shapiro–Wilk normality test; Bilateral EC: *W* = 0.87, *p* = 0.01; Left EC: *W* = 0.89, *p* = 0.03 and Right EC: *W* = 0.81, *p* = 0.001, see "Methods" section), we performed a Wilcoxon signed-ranked test (For completeness, if we apply the Wilcoxon test to the sighted controls data mentioned in the previous paragraph, results did not change: Bonferroni-corrected for multiple comparison across left, right and bilateral hemispheres; Left EC: *V* = 154, *z* = −2.35, *p* = 0.0078, *r* = 0.54, 95% CI = [−0.04, Inf]; Right EC: *V* = 95, *z* = 0, *p* = 0.5, *r* = 0, 95% CI = [−0.15, Inf], all results one-tailed, $\alpha_{Bonferroni}$ = 0.016). We did not find a significant 6-fold symmetry in the early blind participants, neither when considering the bilateral EC ROI (*V* = 77, *z* = −0.7 *p* = 0.76, *r* = 0.16, 95% CI = [−0.39, Inf], one-tailed, α = 0.05) nor the hemispheres separately (Left EC: *V* = 86, *z* = 0.34; *p* = 0.64, *r* = 0.08, 95% CI = [−0.47, Inf], and Right EC: *V* = 104, *z* = −0.34; *p* = 0.37, *r* = 0.08, 95% CI = [−0.27, Inf], all results one-tailed, α = 0.05, Fig. 4E). Notwithstanding the inconsistent pattern of results found in the two groups, grid-like representation in the early blind individuals' left EC was not significantly lower than the one expressed by sighted controls (Wilcoxon ranked-sum test *W* = 231, *z* = −1.06, *p* = 0.14, *r* = 0.24, 95% CI = [−0.14, 0.72], two-tailed, α = 0.05; Fig. 4F). This nonsignificant result might be attributed to the higher individual differences observed within the early blind population (Standard deviation from the mean (SD); sighted controls = 0.21; early blind = 0.96; Levene's test for Homogeneity of variance between groups;

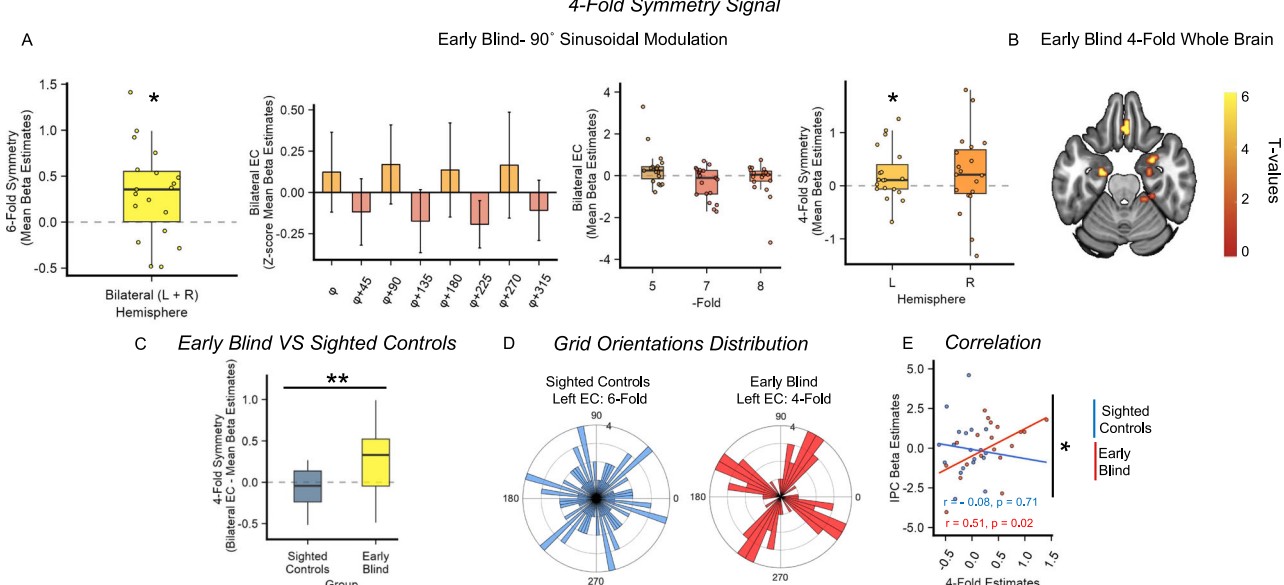

**Fig. 5 | Early blind individuals and sighted showed different neural geometries in the entorhinal cortex. A** Quadrature filter analysis performed on alternative periodicities revealed a significant 90° sinusoidal modulation of the BOLD signal (**A**, right) in early blind individuals' Bilateral EC (one-sample $t$-test; $n = 19$, $p = 0.006$, $\alpha_{Bonferroni} = 0.01$, one-tailed) which can be also visualized as a sinusoid with peaks at 90° and multiples (Data are presented as mean value ± SEM, for display only). The effect is also present although weakened in individual hemispheres (right panel, one-sample $t$-test; $n = 19$, Left EC: $p = 0.038$; Right EC: $p = 0.095$, one-tailed, $\alpha = 0.05$). No other significant modulations of the BOLD signal were detected (middle panel). Source data are provided as a Source Data file. **B** Whole-brain, group-level analyses revealed a cluster of activation in the bilateral EC of early blind participants for the periodicity of interest (90°; The activation overlaps with the MNI-152 T1 template at an uncorrected threshold of $p < 0.01$, One-sample $t$-test), for display only. **C** The 4-fold symmetry effect was significantly higher in early blind group compared to the sighted control group (Two-sample $t$-test; $n = 38$, $p = 0.003$, $\alpha = 0.05$, two-tailed). Source data are provided as a Source Data file. **D** Sighted control 6-fold symmetry left EC grid orientations were uniformly distributed in space (left, Rayleigh test; $n = 19$, $p = 0.55$, $\alpha_{Bonferroni} = 0.01$, two-tailed). Contrarily, the 4-fold symmetry grid orientations in the left EC of early blind individuals were significantly clustered (right, Rayleigh test; $n = 19$, $p = 9.0831e{-}04$, $\alpha_{Bonferroni} = 0.01$, two-tailed). Source data are provided as a Source Data file. **E** Inferior Parietal Cortex (IPC) activity during navigation in the Clock Navigation experiment significantly correlates with the magnitude of the 4-fold symmetry effect in early blind individuals, ($n = 19$, $r = 0.51$, $p = 0.02$, two-tailed), but not in sighted control participants with a significant interaction between the IPC activity and groups ($n = 38$, $p = 0.02$, two-tailed, left panel). Source data are provided as a Source Data file.

$F(1,36) = 11.69$, $p = 0.001$, two-tailed, $\alpha = 0.05$. However, the lack of hexadirectional coding in early blind individuals was strengthened by Bayesian analysis showing no evidence of six-fold symmetry in this group (Left EC $BF_{10} = 0.23$; error % = 0.016; Right EC: $BF_{10} = 0.24$, error % = 0.016; Bilateral EC: $BF_{10} = 0.27$, error % = 0.017).

Accounting for the possible difference in clock-navigation accuracy between the two groups (see above), we had furthermore investigated whether the reduced grid-like activity in early blind participants' EC was attributable to this putative difference. We did not detect any significant correlation between accuracy during the Clock-Navigation experiment and the magnitude of 6-Fold symmetry estimates, not when combining the two groups ($r = -0.16$, $p = 0.32$, $r^2 = 0.02$, 95% CI = [−0.46, 0.16]) nor when we considered the two group separately (sighted controls: $r = -0.18$, $p = 0.44$, $r^2 = 0.03$, 95% CI = [−0.59, 0.29]; early blind: $r = -0.30$, $p = 0.2$, $r^2 = 0.09$, 95% CI = [−0.66, 0.17], Fig. S4). If something, the correlation between 6-fold symmetry estimates and accuracy in early blind individuals showed the opposite (negative) trend, letting us conclude that the reduction of grid-like coding in early blind individuals' EC could be unlikely explained by differences in accuracies between the groups.

In sum, a significant six-fold symmetry emerged in sighted controls (as predicted), but not in early blind individuals. Nevertheless, given the lack of a between-group interaction, we can't conclude that the typical hexadirectional grid code is different across sighted controls and early blind people. However, it might be that the unstable six-fold symmetry in early blind participants' EC is related to an alteration of the typical hexadirectional geometry. We've tested this hypothesis in the following paragraph.

## A different neural geometry in the blind individuals' entorhinal cortex

Given the absence of 6-fold symmetry in the EC, we tested whether early blind individuals showed a significant effect in one of the alternative control models. This analysis revealed the emergence of a significant 90° sinusoidal modulation of the BOLD signal (4-fold) in the early blind individuals' bilateral EC (Bonferroni-corrected across five tested periodicities, one-tailed, one-sample $t$-test; $t(18) = 2.77$, $p = 0.006$, Cohen's $d = 0.64$, 95% CI = [0.11, Inf], $\alpha_{Bonferroni} = 0.01$; Fig. 5A, B) and no other significant sinusoidal modulations (Bonferroni-corrected across five tested periodicities, one-tailed, one-sample $t$-test; 5-fold: $t(18) = 1.72$, $p = 0.05$, Cohen's $d = 0.40$, 95% CI = [−0.0009, Inf]; 7-fold: $t(18) = -1.95$, $p = 0.96$, Cohen's $d = 0.45$, 95% CI = [−0.64, Inf]; 8-fold: $t(18) = -0.89$, $p = 0.8$, Cohen's $d = 0.20$, 95% CI = [−0.5, Inf]. Wilcoxon signed-ranked test: 6-fold, $V = 77$, $z = -0.7$ $p = 0.76$, $r = 0.16$, 95% CI = [−0.39, Inf], one-tailed, $\alpha_{Bonferroni} = 0.01$; Fig. 5A). Crucially, the magnitude of 4-fold symmetry in the early blind group was significantly higher than sighted controls group (Two-sample $t$-test; $t(36) = -3.18$, $p = 0.003$, Cohen's $d = 1.03$, two-tailed, $\alpha = 0.05$, 95% CI = [−0.69, −0.15]; Fig. 5C). The 4-fold effect in early blind people was also present, although weakened, when we considered each hemisphere separately (One-sample t-test; Left EC: $t(18) = 1.88$, $p = 0.038$, Cohen's $d = 0.43$, 95% CI = [0.016, Inf]; Right EC: $t(18) = 1.36$, $p = 0.095$, Cohen's $d = 0.31$, 95% CI = [−0.06, Inf], one-tailed, $\alpha = 0.05$, Fig. 5A). We further investigated the emergence of the 4-fold symmetry in early blind individuals, based on related findings in the literature. Entorhinal 4-fold symmetry in humans has already been observed in two studies, in which sighted participants navigated a virtual-reality environment[45,46]. Interestingly, the 4-fold symmetry has been

reported to be accompanied by a clustering of the participants' preferred grid orientation along the major axes of the environment[45], suggesting an external anchoring of the entorhinal map. Thus, we tested whether a similar clustering was taking place for the blind individuals in our experiment, considering that in a circular environment, no clustering is expected[36,37]. First, we looked at the phase distribution in the left EC in sighted controls for 6-fold symmetry, and we indeed found that they were uniformly distributed (Bonferroni-corrected across left, right and bilateral hemispheres, Rayleigh test of Uniformity; $p = 0.55$, two-tailed, $\alpha_{Bonferroni} = 0.01$, Fig. 5D). However, the phases of the 4-fold symmetry in early blind individuals in the same hemisphere were significantly clustered (Bonferroni-corrected across left, right and bilateral hemispheres, Rayleigh test of Uniformity; $p < 0.001$, two-tailed, $\alpha_{Bonferroni} = 0.01$; Fig. 5D). Although this effect was not present in the right hemisphere, the highly significant clustering effect found in the left EC, consistent with the previous report of 4-fold deformation in humans[45], suggests that the emergence of this neural geometry can be related to anchoring behavior on the main axes of the environment.

The disruption of 6-fold symmetry in the EC has been recently related to the adoption of an egocentric perspective during navigation and the increased activity of inferior parietal areas[47]. Since blind individuals tend to adopt mostly egocentric perspectives during navigation, might the emergence of the 4-fold symmetry be related? We explored this possibility by extracting the beta estimates obtained during a univariate whole-brain analysis[48] (Navigation > Rest; see "Methods" section), in the Clock-Navigation experiment, using the inferior parietal cortex cluster obtained during the analysis of the Nav-Math experiment (defined in the contrast ([Navigation > Math] × [early blind > sighted controls]), see above) as an independent ROI. A significant positive correlation emerged between 4-fold symmetry estimates and the activity in the IPC in early blind individuals (Pearson's product-moment correlation; $r = 0.51$, $p = 0.024$, $r^2 = 0.26$, 95% CI = [−0.08, 0.78], Fig. 5E). The same effect did not emerge in sighted controls ($r = −0.089$, $p = 0.71$, $r^2 = 0.007$, 95% CI = [−0.51, 0.38], Fig. 5E). Furthermore, we performed a linear regression analysis to investigate whether this effect was stronger in early blind individuals compared to sighted controls. Indeed, we detected a significant interaction between IPC activity and groups ($t$-value = −2.33, $p = 0.02$, $r^2 = 0.38$, 95% CI = [−0.31, −0.02], two-tailed, see Table S2) strengthening the hypothesis that the use of an egocentric perspective to navigate the environment influenced the neural geometry of entorhinal cognitive maps in early blind people.

Interestingly, the activity of IPC in early blind people also correlated with their accuracy during the Clock-Navigation experiment ($r = 0.46$, $p = 0.048$, $r^2 = 0.21$, 95% CI = [0.005, 0.75]; Partial Correlation 4-fold: $r = 0.53$, $p = 0.02$), whereas this correlation did not emerge in sighted controls ($r = −0.11$, $p = 0.65$, $r^2 = 0.012$, 95% CI = [−0.53, 0.36]). However, in this case, the interaction between IPC activity and Group was only marginal ($t$-value = −1.8, $p = 0.08$, $r^2 = 0.21$, 95% CI = [0.07, 0.004], two-tailed).

In sum, we documented the emergence of a different neural geometry in early blind individuals' entorhinal cortex during imagined navigation. The grid code detected in early blind people had a periodicity of 90° (4-fold) and was absent in the sighted controls. The 4-fold effect, in blind people, was associated with the clustering of the participants' preferred grid orientation along preferred axes of the clock environment and was correlated with univariate activity in the right inferior parietal cortex during navigation.

## Compared to the sighted, early blind individuals rely more on egocentric navigation strategies during everyday activities and performed worse during real-world navigation

In order to assess their everyday navigation strategies and abilities, we administered to sighted and blind participants a self-report questionnaire[49]. The questionnaire allowed us to compute a general Navigation Confidence score as well as the relative use of Survey and Route strategies, associated with Allocentric and Egocentric space representations, respectively[50–55]. The preference to rely more on one of the two strategies was assessed by computing the difference between Route and Survey scores (dRS); a positive value indicates participant preference for Route strategies (see "Methods" section). No difference between the two groups was detected in the expressed confidence in their spatial navigation abilities (Two-sample $t$-test; Navigation Confidence score: $t(36) = −0.77$, $p = 0.44$, Cohen's $d = 0.25$, 95% CI = [−4.36, 1.94], two-tailed; Fig. 6A).

However, when considering the dRS, a significant difference emerged between the two groups (Wilcoxon ranked-sum test: $W = 94$, $z = −2.27$, $p = 0.01$, $r = 0.41$, 95% CI = [−6, −1], two-tailed; Fig. 6B), indicating early blind people preference for Egocentric strategies during everyday navigation.

As a final step, we conducted a behavioral path integration (PI, Fig. 6C, D) experiment to investigate whether the stronger reliance on egocentric frame of reference of early blind individuals could influence their accuracy in constructing mental representation of the environments in which they are situated. Path integration, defined as the ability to rely on idiothetic cues to update one's current position in the environment in relation to the previously computed position[56,57], has been reported to be a sensitive hippocampal-entorhinal based test for allocentric spatial memories[58,59]. We thus performed a Path Integration experiment on the same participants who performed the MRI task. Participants (Blindfolded; 17 sighted controls and 19 early blind individuals) walked along 16 different linear paths with the help of an experimenter (see Fig S5 and Table S3). Each path contained two stopping points, at which participants were asked to estimate the distance and the orientation between their actual position and the starting point (Fig. 6C). PI performance was calculated for each participant by combining their accuracy in estimating distances and orientations (see "Methods" section). Our results showed that, relative to sighted controls, early blind individuals' PI performance was significantly lower (17 sighted controls & 19 early blind; Two-sample $t$-test; $t(34) = 2.10$, $p = 0.04$, Cohen's $d = 0.70$, 95% CI = [−0.0004,0.024], two-tailed, Fig. 6F).

Right inferior parietal cortex activity has recently been associated with the accuracy with which people encode spatial information, especially when environmental settings or experimental tasks enhance the use of egocentric spatial representations[46,47]. In line with this evidence, in the Clock-Navigation experiment, we observed a positive correlation in early blind individuals between right IPC activity and accuracy during imagined navigation (see above). Would IPC activation level also correlate with navigation accuracy in the real world? To answer this, we correlated the beta estimates extracted in the right IPC in the Clock-Navigation experiment with accuracy in the PI experiment. We found a positive correlation between IPC activity and accuracy during the PI task in early blind participants ($r = 0.50$, $p = 0.028$, $r^2 = 0.25$, 95% CI = [0.064, 0.78]; Fig. 6E), which was largely preserved when we controlled for 6-fold symmetry estimates (partial correlation: $r = 0.48$, $p = 0.041$) and 4-fold symmetry estimates (partial correlation: $r = 0.45$, $p = 0.05$). By contrast, no significant correlation was observed in sighted controls between IPC activity and real-word navigation performance (17 sighted controls, $r = 0.031$, $p = 0.91$, $r^2 < 0.001$, 95% CI = [−0.46, 0.50]; Fig. 6E). Comparing the two groups with linear regression analysis we found a marginally significant IPC activity by Group interaction (17 sighted controls & 19 early blind individuals, $t$-value = −1.95, $p = 0.059$, $r^2 = 0.29$ two-tailed, 95% CI = [−0.02, 0.0005], Fig. 6E).

Although correlations performed with a limited number of subjects should be taken with caution, these results may be considered as preliminary evidence that an increased activity in the inferior parietal cortex, in early blind people, is associated with increased navigation

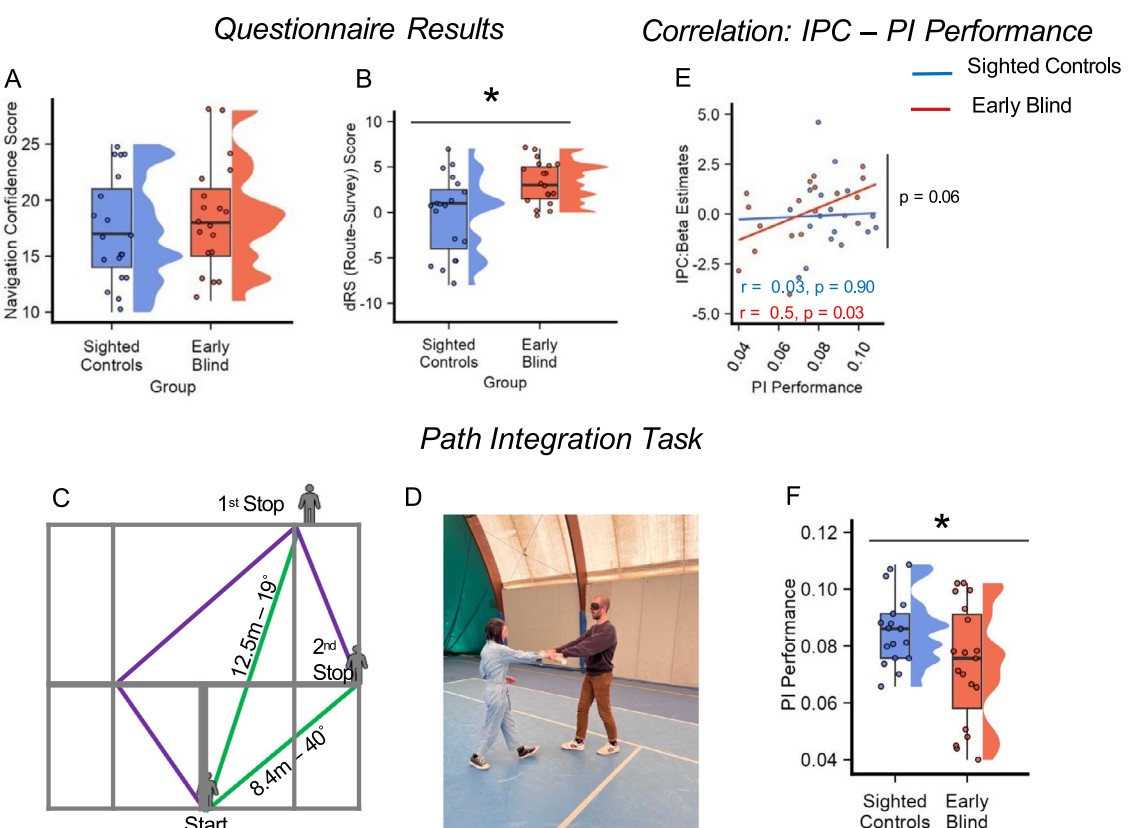

**Fig. 6 | Compared to sighted controls, early blind individuals relied more on egocentric navigation strategies during everyday life and performed worse in a path integration task.** **A**, **B** Participants were asked to fill out a self-report questionnaire on their navigation abilities during everyday life. No differences between groups were detected concerning confidence in spatial navigation abilities (**A**). However, when differentiating between Route and Survey knowledge, we observed a significant difference between the two groups, with early blind participants reporting a greater reliance on Route strategies than sighted controls (**B**, Wilcoxon ranked-sum test; $n = 36$, $W = 94$, $p = 0.01$, two-tailed). Source data are provided as a Source Data file. **C** Path integration task: Half of a tennis court ($10\,m \times 11\,m$) was used to perform the experiment. The experimenter led the blindfolded participants through several paths (purple lines), created using tennis court lines as a reference. Each path contained two stopping points, at which participants were required to estimate the distance and orientation between their actual position and the starting position (green dashed lines). **D** During the path integration experiment blindfolded participants were led by the experimenter through the different paths as

shown in the picture. **E** Inferior parietal cortex activity positively correlated with PI Performance in early blind participants (Pearson correlation; $n = 19$, $r = 0.5$, $p = 0.03$) but not sighted controls (Pearson Correlation; $n = 17$, $r = 0.03$, $p = 0.90$). Furthermore, linear regression analysis detected a marginally significant interaction between IPC activity and accuracy during PI experiment ($n = 36$, $t$-value $= -1.95$, $p = 0.059$, $r^2 = 0.29$ two-tailed). Source data are provided as a Source Data file. **F** Sighted controls were overall more accurate, during real-world navigation, compared to early blind participants (Two-sample $t$-test; $n = 36$, $t(34) = 2.10$, $p = 0.04$, two-tailed). Source data are provided as a Source Data file. In box plots, the boxes indicate the IQR(data points included between the first quartile, 25th percentile, and the third quartile, 75th percentile), the horizontal black line the median (50th percentile), and the whiskers the distance between the first and third quartile to highest and the lowest value in the sample. The distribution of the data is represented by a density curve beside each box. Significance levels are defined as follows: *$p < 0.05$; **$p < 0.01$; ***$p < 0.001$.

performances both in imagined and real-life navigation. Additional correlation analysis between PI performance and the strength of 4-fold and 6-fold symmetry in both groups did not reveal significant results (all $p$-values $> 0.33$), except for a negative correlation between 6-fold symmetry and PI performance in early blind individuals that was, however, clearly driven by outliers and therefore not interpreted (19 early blind individuals; $r = -0.5$, $p = 0.03$, $r^2 = 0.25$, see supplementary materials and Fig. S6).

## Discussion

We investigated the influence of early visual deprivation on the development of the human navigation network (HNN) as well as the development of grid-like coding in the hippocampal-entorhinal system. Here, we discuss the two main findings.

First, the HNN was reliably and selectively activated in both sighted controls and early blind individuals during imagined navigation. Results of the Nav-Math experiment indicate that both groups have greater activation during navigation compared to a mathematics

task in the MFG, PPA, SPL/Precu, and OPA. We did not find navigation-specific activity in the hippocampus, in either sighted controls or early blind participants. This, however, can be explained by the familiarity of both groups with the clock environment, which would not have demanded a strong reliance on episodic memory to retrieve its spatial configuration[60]. Indeed, several studies have shown that hippocampal activity decreases during the exploration or recall of a familiar environment, compared to novel ones[61,62].

Two previous studies have tested sighted controls and early blind people performing spatial navigation during fMRI using a sensory substitution device based on tongue stimulation[31] or the haptic navigation of a 3D maze[32]. In both cases, early blind people activated part of the HNN consisting of PPA, MFG, and SPL/Precu. Activity in the parahippocampal regions was stronger in early blind individuals than in sighted controls, and in general, early blind individuals had higher occipital activations (lingual gyrus, cuneus, fusiform gyrus) than matched sighted controls. By contrast, we did not find any difference between groups in the HNN ROIs (including PPA), and occipital regions

were not specifically active during navigation in either sighted controls or early blind participants (except for OPA). This discrepancy might be due to the fact that, in previous studies, the navigation task was contrasted either with resting periods[32] or scrambled tongue stimulation[31], which requires lower degrees of cognitive control. Here, using a demanding matched-control task with the exact same stimuli (numbers) but requiring a different, non-spatial computation (mathematics), we did not find significantly greater activity in the ventral occipital cortex of early blind individuals. Crucially, when comparing navigation vs. rest, we could see that the occipital cortex was more active in the early blind group than in sighted controls (See Fig. S7), in line with previous reports[31,32]. Nevertheless, such activation cannot be interpreted as strictly navigation-relevant. The only brain region that was differentially and specifically activated during navigation between sighted and blind individuals was the right inferior parietal cortex (IPC), a region of interest known to be important for egocentric spatial representations[44].

This brings us to our second main finding: Early blindness restructured the geometry of entorhinal cognitive maps during the Clock-Navigation task. Our results show that, whereas 6-fold symmetry was detected in sighted controls' entorhinal cortex, replicating previous imagined navigation results[40,41], we did not detect a similar 60° modulation of the BOLD signal in early blind individuals' EC. On the contrary, we found the emergence of 90° sinusoidal modulation (4-fold) in early blind individuals' entorhinal cortex (indicative of a square grid), which was not detected in sighted controls. Although we can provide only a partial explanation for this phenomenon, we hypothesize that 4-fold symmetry derives from the extensive use of an egocentric perspective by blind people during navigation. Egocentric coding of the clock space would increase the reliance on the main axis of the clock, modulating the geometry of grid-cell firing fields and giving rise to a 4-fold symmetry.

The emergence of 4-fold symmetry in the human EC was recently reported in two fMRI studies[45,46] and was generally associated with an anchoring toward the cardinal axes of the environment (north-south; east-west). He and colleagues[45] tested participants in a virtual-reality task, in which they navigated either in an open field or in a hairpin maze. Reflecting the modulation of grid-cell firing fields when rodents navigate a similar maze[60], a 90° rotational symmetry aligned with the main axis of the maze emerged in humans during virtual maze navigation in fMRI[45]. We found a similar anchoring of the 4-fold entorhinal map of early blind participants, despite the absence of barriers in our clock environment (See Fig. 5D), although in our case, grid orientations clustered with a 30° shift from the clock axes that are canonically considered as the main reference (six-twelve and three-nine). In another experiment, Wagner and colleagues[46] found the emergence of 4-fold symmetry when sighted participants had to retrace the path of an avatar that they observed navigating in the previous trial. Similar to our clock experiment, the environment was circular and without barriers. The authors attributed 4-fold symmetry to the possibility that people adopted cardinal axes as a main reference system allowing to compare movement trajectories with these major axes[46]. Although the emergence of the entorhinal 4-fold symmetry could, in principle, be due to different mechanisms in these different experiments, a common factor could be the adoption of an egocentric reference frame during navigation enhanced by the impossibility to collect all the environmental information at a glance (due to the presence of barriers[45]) and by the replication of previously seen movements from a first-person perspective[46]. Interestingly, there is evidence that enhancing participants' self-perception and body awareness during navigation decreases the stability of 6-fold symmetry in the entorhinal cortex and increases activity in the right IPC[47], which, notably, is involved in egocentric spatial coding[42–44]. Moreover, in the experiment by Wagner and colleagues[46], the right IPC activity during the observation of the moving avatar predicted the accuracy during the subsequent path

retracing. Thus, it appears that right IPC activity is associated with both distorted entorhinal geometry and navigation accuracy in tasks that encourage egocentric spatial processing[46,47]. This state of affairs is in keeping with our analysis showing that right IPC activity correlates, in early blind individuals, with the strength of the 4-fold symmetry, and with preliminary evidence associating IPC activity with navigation accuracy in imagined and real-life navigation.

As it emerges from our surveys (Fig. 6B), early blind people relied more on an egocentric coordinate system during everyday navigation[28,30]. The reconstruction of the environment layout in blind people (who cannot perceive several locations at a glance) is necessarily more diachronic and sequential than in sighted people[27]. This enhances the reliance on a first-person perspective and on route-based knowledge grounded in the simulation of successive turns[27,63], which can be easily labeled in memory as "left", "right", "front" and "back" turns, thus with a 90° periodicity. Moreover, it has been shown that egocentric coding increases the dependency on the main axes of the environment to establish item positions and moving directions, which are computed as an egocentric bearing from a given axis aligned with the "point of view" from which the environment is typically experienced[64–68]. The navigation strategy reported by most blind participants (see Table S4) – rotating the clock in order to have the starting/ending point in front of them – is indeed suggestive of a calculation of the direction of movement as a function of egocentric rotation (bearing) from the main sagittal axis (six-twelve) from which clocks are usually experienced. If the direction of movement is calculated by egocentric bearing from the main axes, this axis and the perpendicular one become crucial for orientation, which may re-shape grid-cell firing fields[69], giving rise to 90° rotational symmetry.

In this paper, we have shown that the human navigation network is largely resilient to early visual deprivation. Both early blind individuals and sighted people selectively activate the same set of regions during imagined navigation (compared to an arithmetic task), including occipital regions such as the OPA, usually considered to be highly visual[70–72]. However, we also showed different neural geometries in the entorhinal cortex of sighted and early blind people, with the typical 6-fold symmetry emerging in sighted controls and 4-fold symmetry in early blind individuals. Notwithstanding that the exact mechanisms remain unknown, we report evidence suggesting a relationship between the emergence of 4-fold symmetry and the engagement of the parietal cortex during navigation, together with the anchoring of the entorhinal map to the main axes of the environment. It remains possible that our results could, in part, be related to the specifics of the imagined clock environment, and further studies using different types of environments, as well as auditory or tactile navigation[63] (in which navigation behavior can be better controlled) are needed to fully understand the differences between sighted and early blind individuals during spatial navigation. However, with this study, we show how early blindness can modulate the neural geometry of entorhinal grid maps, possibly by encouraging an egocentric perspective during navigation, shedding light on the general mechanisms underlying the construction of cognitive maps in the entorhinal cortex and providing an initial insight on the consequences of early blindness on their development. Indeed, one still open and fascinating question is whether differences across sighted and blind people would emerge also during conceptual navigation in non-spatial domains[73–76] of knowledge, across complementary egocentric and allocentric reference frames[42,76].

## Methods

### Subjects

Thirty-eight participants took part in the MRI experiments. Nineteen participants were early blind individuals (10 females; age: $M = 37.37$, SD = 6.13; two ambidextrous), and 19 participants constituted the sighted control group (10 females; age: $M = 36.21$, SD = 6.44; one

left-handed). Early blind participants were matched overall by age and sex with sighted control participants, with no significant age difference between the two groups (Paired sample $t$-test; $t(18) = -1.63$, $p = 0.12$). Thirty-six participants took part in the path integration experiment (two sighted control participants did not come back to complete the experiment: 19 early blind individuals & 17 sighted controls, 8 females; age: $M = 35.88$, SD = 6.41; one left-handed). Nonetheless, no difference in age between the two groups was detected (Two-sample $t$-test; $t(34) = -0.70$, $p = 0.48$). Early blind participants were blind since birth or completely lost vision before age 5, reporting, at most, faint light perception and no visual memories. All participants speak Italian fluently, and none reported a history of neurological or psychiatric disorders. The limited sample size was mostly a consequence to the difficulty to find suitable subjects given the rarity of the condition and our strict inclusion criteria. Although this factor should be taken into consideration and future more statistically powerful studies are needed (see also the "Discussion" section), our sample size is comparable to previous studies testing early blind population in spatial navigation and other tasks[31,32,77], as well as to studies testing grid-like coding in virtual/visual[37,78], imagined[40] and conceptual navigation[74]. Moreover, a previous power analysis on studies reporting Hexadirectional coding suggests that a sample size of 20 would have been enough to obtain a 90 power[74]. The ethical committee of the University of Trento approved this study; all participants signed informed consent prior to the experiment and were compensated for their participation. None of the participants was excluded due to excessive head motion (i.e., the maximum head motion of all runs was no more than 3 mm in translation or 3° in rotation).

Supplementary Table 5 shows the demographic information of the early blind.

## Clock-Navigation experiment stimuli

Experimental paths were designed using AutoCAD (v2019), calculating the angle created by each line that connected each number on the clock, represented by the vertex of a dodecagon, to all the others. Repeated paths (e.g., from 9 to 12 and from 12 to 9), paths connecting two adjacent numbers (e.g., from 1 to 2), and those that traverse the center of the space were discarded, resulting in a set of 36 unique paths. Twenty-four path combinations were chosen, ensuring that all the degrees ranging from 15° to 360°, in steps of 15°, were represented, and each number from 1 to 12 was used as starting or ending location an equal number of times (four times per number). We computed the length of each path as the difference between the starting and the target numbers (e.g., from 9 to 12: $9 - 12 = 3$), resulting in three different path lengths: (i) Short (distance between numbers: 3); (ii) Medium (distance between numbers: 4) and (iii) Long (distance between numbers: 5) and balanced it within each run so as to have six short paths; six long paths and 12 medium paths.

Instructions were delivered auditorily to participants (48,000 Hz, 32-bit, mono), and each number was recorded in both a feminine voice and a masculine voice using an online speech synthesizer (https://ttsfree.com/text-to-speech/italian). The average intensity of all the auditory words was thresholded and equalized at 60 dB using Praat 6.1.01 (http://www.fon.hum.uva.nl/praat/).

## Clock-Navigation experiment procedure

Participants performed an imagined navigation task in the fMRI scanner. The task was designed similarly to a previous imagined spatial navigation paradigm[40,41]. Participants were asked to imagine themselves within a clock-like environment, with numbers positioned exactly as on a real clock, and to imagine walking from one number to another, according to instructions. Before the experiment began, we instructed participants to perform the shortest and straightest path possible to arrive at the endpoint, avoiding walking along the perimeter of the clock or stopping or turning in the center. Participants did not receive any instruction concerning the size of the clock or the speed with which they should have had navigated through it, however, they were aware that the imagination period could last a maximum of four seconds. Each trial concluded with a question about the position of a target number in the space as compared to the participants' imagined position at that moment, as shown in Fig. 1A. Prior to the fMRI session, participants performed behavioral training (4 blocks, 96 trials) to familiarize themselves with the task.

The behavioral training was conducted until participants reached a performance equal to or greater than 80% for two blocks consecutively. The fMRI session consisted of 8 runs of 24 trials each (192 trials). In half of the runs, the chosen path combination (see above) was presented in the standard configuration (namely, "Forward" runs). For the other half, participants were guided in the opposite direction ("Backward" runs; e.g., Forward runs: from 8 to 2 – Reverse runs: from 2 to 8). Forward and backward runs were presented interleaved.

Participants heard two numbers (0.5 s each, separated by 0.5 s), recited by a female voice, that corresponded to the starting and ending points. After the ending point instruction, they were asked to begin imagining walking across the clock for a variable amount of time between 4 and 6 s, at the end of which, a third number (target) was recited (0.5 s) by a male voice. Participants were asked to decide whether the target number was situated to the left or the right in relation to the arrival position (4.5 – 6.5 s). The press of a button with the right-hand middle finger indicated the number position was on the left; with the right-hand index finger to indicate the number was situated on the right. Participants were trained to answer before a sound cue played 2 s after the end of the target number instruction. This condition was introduced to push participants to navigate the environment during the imagination time window, reducing the possibility that they start imagining only after having heard the target number. The experimental paths were arranged so that for two consecutive trials, the previous ending point corresponded to the next trial's starting point; when this condition was violated, participants heard a "jump" (In Italian, "salto", 0.5 s) instruction, which indicated that the starting point of the new trial would be different from the ending point of the previous one. They were given 4 s after the jump instruction to reorient themselves according to the new starting position. The target numbers associated with each trial were pseudo-randomized across runs while making sure that they were different from the numbers used as starting and ending positions and were not adjacent to the ending point. Moreover, the positions of the target numbers in space were counterbalanced within each run so that they appeared an equal number of times (12) on the left and on the right side of the space. Target numbers were also counterbalanced across runs in such a way that, for the same trajectory, in half of the runs, the target number was situated on the left and the other half of the runs on the right.

During the entire duration of the experiment, participants were blindfolded and received instruction through MRI-compatible earphones using Psychtoolbox 3.0.14 (http://psychtoolbox.org/). Answers were given using an MRI-compatible response box connected to the testing PC. Stimuli presentation in the behavioral training, as well as the fMRI experiment, were displayed using Matlab releases 2014b (Behavioral Training) and 2017b (fMRI).

## Nav-Math experiment stimuli

Pairs of numbers (i.e., paths for the navigation blocks) were chosen to be identical between math and navigation tasks.

Only the order of presentation of each combination was shuffled across tasks and blocks. Twenty combinations were pseudo-randomly selected out of the 36 available (see above) to make sure that each path was presented only once and not repeated backward. Each number between 1 and 12 was presented at least once and a maximum of six times to avoid number oversampling. Space resolution was not taken

into account in the construction of the navigation blocks, as it was not a crucial feature for univariate analysis contrasting navigation and mathematics. Stimuli recording criteria were the same as those used for the main experiment (see above).

## Nav-Math experiment procedure

During the Nav-Math experiment, participants were asked to complete a navigation task and a mathematic task. The navigation task was similar to the one in the Clock-Navigation experiment (see above). Participants heard three different numbers consecutively, referring to a starting point, an ending point, and a target number, respectively. Each number was played for 0.5 s, interleaved by 0.25 s of silence. Right after the third number was recited, participants had 3 s to decide whether the target number was on the left or on the right of the implied trajectory. Participants were required to answer as quickly and accurately as possible by pressing two keys using the index and middle fingers of the right hand.

At the beginning of the mathematics task, participants heard a target number (0.5 s) for the entire block. Then, each trial in the block had the same structure as in the navigation task. In each trial, three numbers were played (0.5 s each, separated by silence of 0.25 s). Participants had to sum the first two numbers, subtract the third number from the sum, and then decide within 3 s whether the resulting absolute value was greater or smaller than the target number they had heard at the beginning of the block. Participants pressed buttons with their right-hand index finger if the result of the arithmetic operation was less than the target number and with their right-hand middle finger if it was greater. Left/Right and Greater/Smaller responses were counterbalanced within each block (10 times each condition). In both tasks, the first two numbers were recited in a feminine voice, whereas the third number was recited in a masighted controlsuline voice. The Nav-Math experiment consisted of 4 blocks (2 navigation blocks and 2 mathematic blocks) of 20 trials each, interleaved, and counterbalanced across participants. Prior to each block, instructions were played as to which task would be performed. There were 15 s of silence between blocks, allowing the BOLD signal to decay.

Blindfolded participants underwent two runs of the Nav-Math experiment (80 trials for each task). Before participating in the MRI session, all participants performed a brief behavioral training (2 blocks for each condition, 40 trials per condition) to be familiarized with the experimental design.

Audio stimuli were delivered with MRI-compatible headphones using Psychtoolbox 3.0.14 (http://psychtoolbox.org/), and button presses were recorded using an MRI-compatible response box. Behavioral training and the fMRI sessions were performed using Matlab releases 2014b (Behavioral) and 2017b (fMRI).

## MRI data collection

Echo-planar images (EPI) were acquired with a 3 T Siemens Prisma scanner using a 64-channel coil. Functional images were acquired with the following parameters: Field of View (FoV) = 200 mm; Voxel Size = $3 \times 3 \times 3$ mm; Number of slices: 66; Time Repetition (TR) = 1000 ms; Time Echo (TE) = 28 ms; Multi-band acceleration (MB) factor = 6 and a flip angle of 59°. Signal loss in the medial temporal lobe region was addressed by tilting the slice of 15° compared to the anterior-posterior commissure line (ACPC, direction: anterior edge of the slice towards the check). Moreover, to avoid slice group interference, we made sure that the ratio between the number of slices and the MB factor was equal to an odd number[79] (66/6 = 11). Gradient-echo Field maps were acquired for distortion correction of the functional images using the following parameters: FoV = 200 mm; Voxel Size = $3 \times 3 \times 3$ mm³; TR: 768 ms; TE = 4.92 and flip angle of 60°. These images were acquired by adding a fat band to avoid the presence of wrap-around artifacts in the images. In addition, a Multi-Echo MPRAGE (MEMPRAGE) sequence was used to acquire a T1-weighted structural image for each participant

with the following parameters: FoV = 256 mm; Voxel Size = $1 \times 1 \times 1$ mm; TR = 2530 ms; TE1 = 1.69 ms; TE2 = 3.55 ms; TE3 = 5.41 ms; TE4 = 7.27 ms and a flip angle of 7°.

## fMRI data preprocessing

Images underwent standard preprocessing procedures using Statistical Parametric Mapping (SPM12 https://www.fil.ion.ucl.ac.uk/spm/software/spm12/) in Matlab 2020a. Field maps were used to calculate the voxel displacement matrix to reduce distortion artifacts. Functional images were spatially realigned and corrected with the voxel displacement matrix using the 'Realign and Unwarp' tool in SPM. Structural images (T1) were realigned to the mean of functional images, and the functional images were normalized to the Montreal Neurological Institute (MNI) space using unified segmentation methods. Images were smoothed with a 6 mm full-width-at-half-maximum (FWHM) spatial kernel. Whole-brain analyses were implemented on the smoothed images in the MNI space; ROI analyses in the EC were implemented on the unsmoothed images in the native space.

## Nav-Math experiment: whole-brain analysis

First, five conditions were modeled in the General Linear Models (GLMs) at the first level: navigation instructions, mathematic instructions, mathematic target number, navigation task, and mathematic task. The duration of the navigation and mathematic tasks was calculated from the onset of the first number until the participants' responses (Purple boxes, Fig. 1B, C). The produced boxcar functions were convolved with the hemodynamic response function (HRF). The six-rigid head motion parameters, computed during the spatial realignment in SPM, were included in the GLMs to control for head motions. Furthermore, slow drifts in the signal were removed using a high-pass filter at 1/256 Hz. Second, we performed a non-parametric permutation analysis using the SnPM toolbox for SPM (http://nisox.org/Software/SnPM13/). One-sample t-test (early blind individuals and sighted controls independently) or two-sample *t*-test (Sighted controls > early blind and early blind > sighted controls) were computed on each participant's beta maps obtained by contrasting the two tasks (Navigation > Math and Math > Navigation) and by contrasting each task with the resting state periods (Navigation > Rest and Math > Rest). An Explicit mask of the main brain areas involved in navigation processing was set in the model. The HNN ROI was obtained by searching for the term 'navigation' in Neurosynth (https://www.neurosynth.org/) which integrated brain activation from 77 studies (cluster of activation thresholded at $p_{FDR} < 0.01$). We ran 10000 permutations, and no variance smoothing was performed. A voxel-level FDR corrected $p = 0.05$ was set as the threshold for multiple comparisons. In addition, group-level analysis using the same data reported above was performed on unmasked images, allowing exploratory whole-brain analysis. Lastly, small volume correction (SVC) analysis was performed on beta maps resulting from the group and task contrast ([Navigation > Math] × [early blind > sighted controls]) to assess possible group difference in the parietal cortex with the voxel-level significant threshold at $p < 0.001$ and the cluster-level significant threshold at $p_{FWE} < 0.05$.

## ROI definition

Subject-specific entorhinal cortex in the left and right hemispheres were cytoarchitectural defined and converted from surface space to volume space using the 'mri_convert' function in Freesurfer (v7.1.1, https://surfer.nmr.mgh.harvard.edu/). The obtained masks were co-registered to the mean functional image to obtain the same spatial resolution ($3 \times 3 \times 3$ m³). On average, sighted controls' left EC included 116 voxels and right EC 96 voxels, and early blind individuals' left EC included 106 voxels and right EC 93 voxels. Analyses were conducted both using individual hemisphere EC masks and bilateral EC masks combining both hemispheres[37].

## Clock Navigation experiment: grid-like signal analyses

Four-way cross-validation has been applied to quadrature filter analysis previously used to detect grid-like coding in humans' EC during spatial navigation tasks[36,40,78]. The eight experimental runs were divided into 4 partitions of 2 runs each, combining a 'forward' and a 'backward' run (see above). For each iteration, three partitions (6 runs) were used in GLM-1 to estimate the preferred grid orientation, and the remaining partition (2 runs) was used in GLM-2 to test the strength of the grid-like representation in the EC. The process was iterated until each partition was used for GLM-1 (Estimate) three times and for GLM-2 (Test) once (4 iterations). Within each partition, trajectories' degrees were equally represented. In both GLMs, the signal was high-pass filtered at a threshold of 1/128 Hz to remove the slow drift, and the six-rigid head motion parameters computed during the spatial realignment in SPM were modeled as nuisance regressors. Both GLMs also included two regressors of no interest: the 'jump instruction' period and the period computed from the onset of the 'target number instruction' event until participants' responses.

More specifically, GLM-1 was first used to estimate the preferred grid orientation for each participant in their native space. Imagined navigation periods (purple box, Fig. 1A) were modulated by two parametric regressors: $cos(6\theta_t)$ and $sin(6\theta_t)$, where $\theta_t$ indicates the trajectory's angle at trial 't.' The factor '6' indicates that six-fold symmetry rotational periodicity was tested. Resulting beta maps, $\beta_1$ and $\beta_2$, were used to calculate each voxel's grid orientation as:

$$\varphi = \left( arctan \left[ \frac{\beta_2}{\beta_1} \right] \right) / 6 \tag{1}$$

Participants' preferred grid orientation was calculated as a weighted mean across all the voxels within the EC ROI. The weight of the mean reflected the amplitude of the voxel response calculated as follows:

$$\sqrt{\beta_2^2 + \beta_1^2} \tag{2}$$

Second, the remaining partition was used in the GLM-2 to test the emergence of a 60° sinusoidal modulation of the BOLD by calculating the cosine of the trajectories' angle performed by participants realigned with their preferred grid orientation as in:

$$cos(6(\theta_t - \varphi)) \tag{3}$$

The magnitude of grid-like representation in each participant was quantified by averaging the parametric regressors estimated in the GLM-2 across all the iterations. Finally, to better visualize the effect, we divided the trajectories' angles into 12 bins aligned with each participant's mean grid orientation (60° modulo 0) or misaligned (60° modulo 30) and averaged the corresponding regressors across participants (Fig. 4B middle). All the analyses were conducted on the periodicity of interest (6-fold) as well as on control models (4-, 5-, 7-, 8- Fold symmetries).

Finally, we modeled a group-level GLM to conduct an exploratory whole-brain analysis. One-sample *t*-test (independently for sighted controls and early blind individuals) was conducted on beta maps obtained in the GLM-2 to investigate which brain areas were sensible to 6-fold symmetry.

## Path integration experiment procedure

The path integration behavioral experiment was constructed following[78] experimental design. Differently from the classical path integration tasks, where participants were asked either to estimate the distance or orientation of a target point from a single location, this experimental design required participants to both estimate distance and orientation of a target point from two different locations in the space. The use of multiple locations enabled the collection of a greater amount of data points, which would provide a more accurate calculation of the PI errors. Moreover, distance and orientation measurements acquired from different points in space might reduce the biases in the estimations that could arise when participants need to provide a single answer at the end of the walked path[78]. Here, we used half of a tennis field (10.97 · 11.88 m) to construct eight unique paths following the field's predefined lines (Fig. S3). In Each path, two stopping points were introduced with distances from the starting location ranging from 4 m to 14 m and angles from 37° to 122°.

Participants were blindfolded before the entrance into the experimental environment. They were provided with a compass attached to a string around their necks and a cardboard stick in their hands during navigation. Before the experiment started, they received verbal instructions about the task, and once they confirmed to have understood all the instructions, the experiment started. During the experiment, participants placed both hands on the stick. The experimenter stood in front of them, with a hand on the cardboard stick, and led them to walk along the path. Once reached the first stopping point, the experimenter asked participants to first estimate in meters and centimeters the Euclidean distance between their actual position and the starting point and then to point toward the direction of the starting position using the compass. The answers were transcribed by the experimenter. Participants completed eight paths, repeated twice (16 paths), and none of them reported to have recognized the repeated paths. At the start and the end of the experiment, participants completed two standardization paths used in the analyses to correct their distance estimations (see the "Path integration analyses" section). They were asked to walk two straight paths of 5 m and 10 m and to guess the walking distance in meters. Moreover, at the end of the experiment, they were asked whether they had heard any external cues that helped them to orient in the space during the task, but this was not the case for any of the participants.

Supplementary Table 2 reports the list of the performed paths with their associated measurements.

## Path integration analyses

As for the experimental design, analyses were conducted following[78]. Participants' distance estimations were corrected using the standardization paths data to reduce the biases in the computation of the error not directly ascribable to the participants' path integration ability but rather to their tendency to underestimate or overestimate distances in real life. First, the participants' answers obtained at the beginning and the end of the experiment in the 'standardization paths' were averaged for each specific distance, 5 m and 10 m. Subsequently, we calculated the correction factor ($C_f$) for both distances as follows:

$$C_f = d_{standardize} / d_{response} \tag{4}$$

Where $d_{standardize}$ was the actual length of the path, and $d_{response}$ was the participants' answer. Second, the computed correction factors were multiplied by the participants' estimated distances during the task. Reported distances from 4 m to 7.5 m were corrected using the 5 m $C_f$, and those greater than 7.5 m using the 10 m $C_f$. Corrected distances ($d_{corrected}$) and orientations (Ori) estimations of each stopping point were combined to calculate the x and y coordinates of the presumed starting position:

$$x_{presumedStart} = x_{stop} + d_{corrected} * \cos \cos(Ori)$$
$$y_{presumedStart} = y_{stop} + d_{corrected} * \sin(Ori) \tag{5}$$

Where $x_{real}$ and $y_{real}$ corresponded to the real coordinates of the starting position of each trial. In order to compute the path integration error of each stopping point, the Euclidean distance between the stopping point and the presumed starting point estimated at the previous stopping point was calculated (PI Error). For the first stopping

point, the previous presumed starting position corresponded to the real starting position of the path. This procedure allowed us to calculate PI independently for each path segment, reducing the possible biases produced by a cumulative computation of it. Lastly, the calculated errors were averaged across stopping points and trials, and the performance was computed as follows:

$$PI_{performance} = \frac{1}{PI_{error}} \qquad (6)$$

### Questionnaires

Participants were asked to fill out two questionnaires after the fMRI experiment: one self-made questionnaire investigating which strategies they were using to navigate within the clock environment (See Supplementary Table 4) and a self-report questionnaire aimed to investigate participants' navigation strategies in their everyday life[49]. The scores from the QOS were obtained by summing the points of specific questions following the guidelines provided by the authors[49] (Navigation Confidence: question 3+ question 4+ question 5 s+ question 10+ question 11+ question 13; Route knowledge: q5r + q6r; Survey knowledge: q5s + q6s + q9$_{sp}$ − 9$_{verb}$). The dRS scores were computed by subtracting the Survey knowledge score from the Route knowledge score so that a positive value would indicate participants' tendency to rely on more egocentric strategies during their everyday life navigation, and negative values would have indicated the opposite.

### Statistical analyses

Statistical analyses were conducted using SPM and SnPM in Matlab 2020a for whole-brain analyses and R (v4.2.2) for ROI analyses. The distributions of the grid orientations (Fig. 4D) were assessed using the function in the CircStat toolbox in Matlab 2020a. Results of both the whole-brain and the ROI analysis were computed using one-sample t-test to investigate single-group effects and a two-sample t-test to investigate the effect between groups. The Wilcoxon Signed-Rank test was used if the data obtained from the ROI analyses were not normally distributed. The normal distribution of the estimated parameters was checked using the Shapiro–Wilk test.

We computed Pearson's product-moment correlations to assess the relationship between parietal cortex activity and fold symmetries or path integration performance. The correlations with brain activity in the EC were conducted using the 4-fold beta estimates in the bilateral EC of both sighted controls and early blind individuals, whereas the correlations with 6-fold symmetry were conducted on the left EC beta estimates for sighted controls and the bilateral EC beta estimates for the early blind individuals. The significance threshold was set at α < 0.05 and the p-value computed two-tailed, unless specified. For each statistical test, effect sizes were computed using R (v4.2.2). The behavioral and the ROI results were represented using box-plot and raincloud plots, where the boxes indicate the interquartile Range (IQR), meaning the data points included between quartile 1 (Q1, lower quartile, 25th percentile) to quartile 3 (Q3, upper quartile, 75th percentile), the horizontal black line indicates the median of the values of the sample, and the whiskers indicate the distance between the upper and lower quartile to highest and the lowest value in the sample. Moreover, in raincloud plots, the distribution of the data is represented by a density curve beside each box. The sample size for all the analyses was of 19 participants in the sighted control group and 19 participants in the early blind group unless differently specified. Asterisks upon the graphs represent significance level as follow: *$p$ < 0.05; **$p$ < 0.01; ***$p$ < 0.001.

### Reporting summary

Further information on research design is available in the Nature Portfolio Reporting Summary linked to this article.

## Data availability

The neuroimaging and behavioral data generated in this study have been deposited in the Zenodo database under accession code https://doi.org/10.5281/zenodo.10794697[80]. Source data are provided with this paper and are available under accession code https://doi.org/10.5281/zenodo.10794697[80]. The raw and pre-processed neuroimaging data are protected and are not available due to data privacy laws, access can be obtained by forwarding a formal inquiry to the corresponding author (federica.sigismondi@unitn.it). Source data are provided with this paper.

## Code availability

The codes to generating the main figures and for the behavioural task have been deposited in the Zenodo database under accession code https://doi.org/10.5281/zenodo.10694439[80].

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

## Acknowledgements

This work was supported by the European Research Council grant (ERC-StG NOAM – 804422 awarded to R.B.). We are thankful to our blind and sighted participants for their collaboration. We are furthermore grateful to Jorge Jovicich, Nicola Pace, Stefano Tambalo, Manuela Orsini, and Ilaria Mirandola for technical assistance in developing fMRI acquisition sequences. Finally, we thank all the members of the BottiniLab for the insightful discussion about this project.

## Author contributions

Conceptualization: F.S., Y.X., R.B., M.S. Methodology: F.S., Y.X., R.B., Investigation: F.S., M.S., R.B. Formal analysis: F.S., Y.X., R.B. Supervision: R.B. Writing – original draft: F.S., Y.X., R.B. Data curation: F.S.

## Competing interests

The authors declare no competing interests.
