## [Peer Review File · Nature Communications]

Altered grid-like coding in early blind peopleReviewer #1 (Remarks to the Author):

Sigismondi et al present an fMRI and behavioural study in sighted and early blind individuals designed to assess grid-like responses in entorhinal cortex and how they are modified in early blind individuals. They show the predicted 6-fold grid-like signal in sighted controls (SC), but not 6-fold signal in the early blind (EB) group. Interesting, there is a modulation of the BOLD response in entorhinal cortex in the EB group, but it shows a 4-fold signal, consistent with a more square (rather than hexagonal) grid-like structure. They further show this 4-fold signal is aligned across EB participants (but is not in SC participants) and the strength of this 4-fold signal correlates with univariate BOLD response in the inferior parietal cortex (IPC). Finally, they show that EB participants show impairments in a separate path integration task and that performance in this task in the EB participants again correlates with univariate signal in the IPC. They conclude that the "geometry of the entorhinal cognitive map" is altered in EB participants, and that the IPC and path integration results point to a more egocentric strategy relative to SC participants.

The study question is very interesting and theoretically important. A few studies have looked at spatial navigation in EB, however the degree to which EB individuals rely on similar grid cell populations is unclear, and how these signals might link to real-world navigation behaviour has not been studied. I believe this makes the question, methods, and results highly novel and interesting. While I might question some of the conclusions, and in particular the strength of the evidence presented (see comments below), the findings are important and will be of interest to a wide range of scientists.

Major Comments:

1. My biggest concern is the relatively low N (19 per group). The authors do show a predicted 6-fold signal in SC participants, however only in the left EC and not in the right or the combined bilateral analysis. They also present quite a few correlations done in each group, and N=19 for exploratory correlations is not ideal unless the effect size is very large (which is unlikely to be the case here). I do not think this should prevent publication, as I think the study is still very interesting, however I would suggest the authors note this issue and perhaps tone down some of the conclusions as a function of this issue. This study seems like an excellent starting point for future well-powered research.
2. The difference between accuracy in the two groups in the Clock-Navigation experiment isn't significant, however there is a 'borderline' effect ($p=0.052$) that might be worth noting. This is particularly the case as there appears to be a low-performing outlier in the SC group (see Figure 1C). If you remove this outlier is there a significant difference between the groups? Given this section is mostly focussed on the lack of differences between the groups, I wonder if Bayes Factors in favour of the null might strengthen your argument.
3. P. 6 – you show 6-fold signal in the left EC but not right EC (in the SC group), and then show no 4, 5, or 7 fold signal in left EC. For completeness, I think you should also show no 4, 5, or 7 fold in the right EC.
4. I understand why you used a non-parametric statistic to test for 6-fold signal in the EB group, however it does mean the stats you use differs across groups. For completeness, it would be good to show stats across both a parametric and non-parametric test for both groups, so the reader can compare the groups more directly.
5. You show the 4-fold signal in EB is greater than in SC, however there isn't a difference in the 6-fold signal in SC relative to EB. For completeness, you could present the interaction between the two (i.e., $(4\text{-fold EB} - \text{SC}) - (6\text{-fold EB} - \text{SC})$).
6. P. 9 – you present evidence for a correlation between the 4-fold signal and activity in IPC in the EB but not SC groups. It would strengthen your findings to show that there is a significant difference between the slopes of these two correlations.
7. P. 13 – "which would not have demanded a strong reliance on episodic memory to retrieve its

spatial configuration". I find this argument slightly odd, given the strong emphasis of the paper on cognitive maps in the hippocampal-entorhinal system, to then dismiss a lack of hippocampal activation in terms of episodic memory. I sympathise with the argument, but it might need to be articulated a bit more carefully.

Reviewer #2 (Remarks to the Author):

Sigismondi and colleagues investigated entorhinal grid-like codes in early blind people. Given that the hippocampal-entorhinal system is linked to the construction of allocentric spatial-cognitive maps, primarily driven by visual input in humans, the authors asked whether early blind individuals would show altered grid-like codes during (imagined) navigation through a clock space compared to sighted controls.

They found that although both early blind and sighted individuals engaged brain regions typically involved in spatial navigation (Human Navigation Network) to a similar extent, grid-like codes were not significantly increased in early blind individuals. With additional analyses, they showed that inferior parietal activation appears increased during navigation in blind individuals and that this effect is further linked to increased clock navigation performance, path integration, and a 4-fold (rather than a 6-fold) rotational symmetry of the entorhinal BOLD signal (which they discuss as potential evidence for egocentric rather than allocentric navigation strategies in early blindness).

The results are interesting and would certainly have an impact on the field, the paper is well-written. My main comments pertain to the very small (and not significant) difference in (6-fold) grid-like codes between the groups – making it difficult to clearly decide whether grid-like codes are actually altered in early blind compared to sighted individuals.

Main comments:

Clock navigation experiment: Did the participants receive any instruction on how large they should imagine the clock-like space or with what walking speed they should imagine to navigate? For instance, imagining a smaller space would lead to shorter path distances (which could be coupled to lower grid signals). Did participants have to indicate via button press when they completed the trajectory? This was not clear to me from the task description.

Close navigation experiment, accuracy: In fig 3c it seems that there is an outlier in the sighted control group (value between 0.8-0.85%). If so, this would likely contribute to the non-significant difference between SC and EB (in other words, it could be problematic if EB performance is significantly lower than SC after outlier exclusion, as the performance difference could explain the non-significant grid-like codes).

It is difficult to interpret the meaning of significantly increased grid-like codes in SC but not EB without a significant group difference (i.e., the fact that SC is significant but not EB does not necessarily mean that SC > EB, or that grid-like codes in EB are not present). It would be helpful if the authors could confirm the null result using Bayesian statistics to ameliorate some of the concerns related to the very similar grid-like values in SC and EB. This point makes it difficult to state that grid-like codes in EB are truly altered.

Minor comments:

Abstract, line 12, there are two commas “,”

Abstract, line 15, “Human Navigation Network”, it might not be clear to every reader what this entails. I would suggest writing out (at least the core regions).

Introduction, line 61, whether fMRI-based grid-like codes are “a reliable proxy for the activity of grid cells” is debated in the field and I would suggest to phrase this more carefully

Results, please report sample sizes and effect sizes for each analysis step / result

I was not able to find the description in the text belonging to Fig. 3H. I think there is something missing?

Reviewer #3 (Remarks to the Author):

The manuscript describes extensive work on spatial cognitive maps in early blind versus sighted controls. Two types of analysis on fMRI data are combined with behavioural data. The experiments are carefully performed and well executed, and processing of data is fine. The experiments are well described and there is enough detail in the methods to understand how the experiments were performed. The main concern is statistics (see main comment). The main conclusion is that early blind show a 90 degrees rotational symmetry (4-fold) and controls a 60 degrees rotational symmetry. The results for the early blind are significant for both hemispheres, while the results for the controls are only significant for the left hemisphere.

Main point: Bonferroni corrections are applied, but inconsistently, and in an unclear way. In principle, the comparison that is made in the experiment for the main conclusion is for 3 types of analysis (bilateral, left hemisphere and right hemisphere) and for 5 models (4, 5, 6, 7, and 8 folds). This would in principle mean a bonferroni correction for $5 \times 3 = 15$ comparisons, but if this would be applied no statistical differences can be reported for any of the results. In the sighted control condition the bonferroni correction is applied only for hemisphere conditions (3). For the early blind it is applied for four tested periodicities. Why this difference for SC vs EB? Why in the case of EB only four, since there were 5 periodicity models tested? Also the correction is for four cases according to the results section and the p-value is 0.025, why is that called a significant effect? In principle, the fact that results are insignificant is not a problem when there is a very clear hypothesis to start with. For the 6-fold model in sighted controls this might be the case, also because this has been reported before in previous studies. However, the strong conclusion about the 4-fold representation in early blind looks more like a post-hoc finding, without properly correcting for the number of comparisons. By taking this into account the manuscript has to be drastically rewritten.

Minor point: Second sentence in the abstract has two comma's.

REVIEWER COMMENTS

Reviewer #1 (Remarks to the Author):

Sigismondi et al present an fMRI and behavioural study in sighted and early blind individuals designed to assess grid-like responses in entorhinal cortex and how they are modified in early blind individuals. They show the predicted 6-fold grid-like signal in sighted controls (SC), but not 6-fold signal in the early blind (EB) group. Interesting, there is a modulation of the BOLD response in entorhinal cortex in the EB group, but it shows a 4-fold signal, consistent with a more square (rather than hexagonal) grid-like structure. They further show this 4-fold signal is aligned across EB participants (but is not in SC participants) and the strength of this 4-fold signal correlates with univariate BOLD response in the inferior parietal cortex (IPC). Finally, they show that EB participants show impairments in a separate path integration task and that performance in this task in the EB participants again correlates with univariate signal in the IPC. They conclude that the “geometry of the entorhinal cognitive map” is altered in EB participants, and that the IPC and path integration results point to a more egocentric strategy relative to SC participants.

The study question is very interesting and theoretically important. A few studies have looked at spatial navigation in EB, however the degree to which EB individuals rely on similar grid cell populations is unclear, and how these signals might link to real-world navigation behaviour has not been studied. I believe this makes the question, methods, and results highly novel and interesting. While I might question some of the conclusions, and in particular the strength of the evidence presented (see comments below), the findings are important and will be of interest to a wide range of scientists.

Major Comments:

1. My biggest concern is the relatively low N (19 per group). The authors do show a predicted 6-fold signal in SC participants, however only in the left EC and not in the right or the combined bilateral analysis. They also present quite a few correlations done in each group, and N=19 for exploratory correlations is not ideal unless the effect size is very large (which is unlikely to be the case here). I do not think this should prevent publication, as I think the study is still very interesting, however I would suggest the authors note this issue and perhaps tone down some of the conclusions as a function of this issue. This study seems like an excellent starting point for future well-powered research.

We thank the reviewer for the positive comments. We acknowledge the limitation concerning our sample size. Testing special populations like the early blind in MRI is

indeed challenging because of the rarity of the condition and our strict inclusion criteria (e.g., no residual pattern vision, no visual memory, and fMRI compatibility).

In the new version of the manuscript, we clearly acknowledge this limitation (see below) and highlight how our study is a promising starting point for future research in the field. However, at the same time, we would like to highlight that our sample size is comparable to previous studies testing early blind population in spatial navigation and other tasks (Kanjilia et al., 2015; Gagnon et al., 2012; Kupers et al., 2010), as well as to studies testing grid-like coding in virtual/visual (Stangl et al., 2018; Nau et al., 2018), imagined (Horner et al., 2016) and conceptual navigation (Park et al., 2021), some of which correlates brain activity with behavioral performance (Stangl et al., 2018). Moreover, although experimental power might not be the ideal one (because of the constraints mentioned above), a previous study on hexadirectional coding demonstrated that a sample size of 20 would have been enough to obtain a .90 power (Park et al., 2021). Thus, the sample size used in our study is sufficient to detect a reliable signal in the EC and is in line with several influential studies in the field.

We summarize this perspective in the “participants” section of the material and methods (page 18 of the revised manuscript):

“The limited sample size was mostly a consequence of the difficulty in finding suitable subjects given the rarity of the condition and our strict inclusion criteria. Although this factor should be taken into consideration and future more statistically powerful studies are needed (see also the Discussion section), our sample size is comparable to previous studies testing early blind population in spatial navigation and other tasks (Kanjilia et al., 2015; Gagnon et al., 2012; Kupers et al., 2010), as well as to studies testing grid-like coding in virtual/visual (Stangl et al., 2018; Nau et al., 2018), imagined (Horner et al., 2016) and conceptual navigation (Park et al., 2021). Moreover, a previous power analysis on studies reporting hexadirectional coding suggests that a sample size of 20 would have been enough to obtain a .90 power (Park et al., 2021).”

We comment on the interpretation of correlation analysis with a relatively low N at page 15:

“Although correlations performed with a limited number of subjects should be taken with caution, these results may be considered as preliminary evidence that an increased activity in inferior parietal cortex, in the EB, is associated with increased navigation performances both in imagined and real-life navigation.

Additional correlation analysis between PI performance and the strength of 4-fold and 6-fold symmetry in both groups did not reveal significant results (all p-values > 0.33).”

Moreover, we reiterate the need for further studies to confirm and extend our results in the discussion section (page 17):

“It remains possible that our results could, in part, be related to the specifics of the imagined clock environment, and further studies using different types of environments, as well as auditory or tactile navigation (Ottink et al., 2021; in which navigation behavior can be better controlled) are needed to fully understand the differences between sighted and early blind during spatial navigation. However, with this study, we show for the first time how early blindness can modulate the neural geometry of entorhinal grid maps, possibly by encouraging an egocentric perspective during navigation, shedding light on the general mechanisms underlying the construction of cognitive maps in the entorhinal cortex and providing a first insight on the consequences of early blindness on their development. One still open and fascinating question is whether differences across sighted and blind would also emerge during conceptual navigation in non-spatial domains^{72–75} of knowledge, across complementary egocentric and allocentric reference frames (Bottini et al., 2020; Viganò et al., 2023).”

Concerning the lateralization of the six-fold signal in the EC, it is worth noticing that the two studies in the literature that investigate hexadirectional coding during imagined navigation also report lateralized grid-like coding in the left EC (Horner et al., 2016; Bellmund et al., 2016). Although the reasons of such lateralization are not clear, this result in the sighted is not unexpected, and Bonferroni-corrected statistics across hemispheres ensure the strength of our results (see also the reply to Reviewer 3). We’ve included this explanation in the results section (page 8):

“Interestingly, the left lateralization of the grid-like signal was also reported in two previous studies on imagined navigation (Bellmund et al., 2016; Horner et al., 2016).”

2. The difference between accuracy in the two groups in the Clock-Navigation experiment isn’t significant, however there is a ‘borderline’ effect ($p=0.052$) that might be worth noting. This is particularly the case as there appears to be a low-performing outlier in the SC group (see Figure 1C). If you remove this outlier is there a significant difference between the groups? Given this section is mostly focussed on the lack of differences between the groups, I wonder if Bayes Factors in favour of the null might strengthen your argument.

We thank the reviewer for having highlighted the borderline effect resulting from the two-sample t-test on the accuracy between the two groups (EB vs SC). We’ve performed additional analyses on the behavioral data concerning the clock navigation experiment, which indeed revealed an outlier in the performance of the SC group but no outlier in the

EB group. After removing the outlier, we performed once again the two-sample t-test between groups, which showed a significant difference in the accuracy between the two groups ($n = 37$, $t_{(35)} = 2.87$, $p = 0.007$). Similarly, the Bayesian t-test demonstrated that there was very weak evidence in favor of a difference in the accuracy between the two groups ($BF_{10} = 1.237$; error % = 0.006, Kass et al., 1995) considering the full sample (i.e., 19 SC & 19 EB) but substantial evidence when the outlier in the SC group was removed ($BF_{10} = 6.665$; error % = 2.6×10^{-6} , Kass et al., 1995). We noted this at the end of the paragraph in the new version of the manuscript (page 3). However, the average accuracy, in both groups is very high (19 SC, Accuracy = 95%; 19 EB, Accuracy = 92%). Thus, the main point of the paragraph, namely that both sighted and blind can successfully navigate the clock space (as reported in the title), remains intact. The information has been integrated with the paragraph as follows (pages 3-4):

“No difference in accuracy was detected between groups and tasks in the Nav-Math experiment (Repeated-measure ANOVA; main effect of group: $F_{(1,36)} = 0.86$, $p = 0.36$, $\eta^2 = 0.018$; main effect of task: $F_{(1,36)} = 0.85$, $p = 0.36$, $\eta^2 = 0.005$; Group \times Task interaction: $F_{(1,36)} = 1.42$, $p = 0.24$, $\eta^2 = 0.008$, two-tailed; Fig. 1D). Reaction times (RTs) followed a similar pattern with no differences detected in RTs between groups and tasks in the Nav-Math experiment (main effect of group $\chi^2_{(1)} = 1.53$, $p = 0.21$; main effect of task: $\chi^2_{(1)} = 3.59$, $p = 0.06$; Group \times Task interaction: $\chi^2_{(1)} = 0.24$, $p = 0.62$, two-tailed, Fig. 1F). Collectively, these results suggest that both sighted and blind were able to perform the tasks and to perform comparably.

Similarly, in the Clock-Navigation experiment, we did not detect differences in accuracy (two-sample t-test; $t_{(36)} = 2.0$, $p = 0.052$, Cohen's $d = 0.62$, two-tailed, Fig. 1C) and RTs (Linear Mixed-Effect Model; $\chi^2_{(1)} = 2.45$, $p = 0.12$, two-tailed, Fig. 1E) between Groups. However, outlier analysis revealed the presence of one outlier in accuracy in the SC group. Thus, we've performed the analyses excluding the outlier participants. Results on the RTs remain invariant with no differences between groups (Linear Mixed-Effect Model; 18 SC & 19 EB, $\chi^2_{(1)} = 2.28$, $p = 0.13$, two-tailed). On the other hand, analyses on accuracy detected a significant difference between the two groups (18 SC & 19 EB, $t_{(35)} = 2.87$, $p = 0.007$, Cohen's $d = 0.94$, two-tailed). Despite this slight difference in navigation accuracy, it is important to consider that the average accuracy was very high in both groups (19 SC, accuracy % = 95; 19 EB, accuracy % = 92), therefore we could conclude that both sighted and blind participants were able to successfully navigate the clock environment during the Clock-Navigation experiment.”

However, one may wonder whether the lower accuracy in the blind group may be related to the lack of six-fold symmetry (See also Reviewer 2, Point 2). We consider this unlikely. The 6-fold symmetry estimates did not correlate with the accuracy score neither when we combined the two groups (19 SC & 19 EB; $r = -0.16$, $p = 0.32$, $r^2 = 0.02$) nor when we

considered the two groups separately (19 SC: $r = -0.18$, $p = 0.44$, $r^2 = 0.03$; 19 EB: $r = -0.30$, $p = 0.2$, $r^2 = 0.09$, see figure below). If something, the correlation between 6-fold symmetry estimates and accuracy in EB individuals showed the opposite (negative) trend, suggesting that participants who had better performance were also those with a lower expression of the 6-fold symmetry in EC. These results remained stable also when we removed the outlier participants in the SC group (Overall, 18 SC & 19 EB; $r = -0.17$, $p = 0.3$, $r^2 = 0.09$; Single groups: 18 SC, $r = -0.34$, $p = 0.16$, $r^2 = 0.11$; 19 EB, $r = -0.30$, $p = 0.2$, $r^2 = 0.09$).

We report these analyses and the above figure in the Supplementary material, and we refer to it at the end of the section: “**Six-fold grid-like coding did not emerge in early blind**” (Page 8-9):

“Accounting for the possible difference in clock-navigation accuracy between the two groups (see above), we’d furthermore investigated whether the reduced grid-like activity in EB participants’ EC was attributable to this putative difference. We did not detect any significant correlation between accuracy during the Clock-Navigation experiment and the magnitude of 6-Fold symmetry estimates not when combining the two groups (19 SC & 19 EB, $r = -0.16$, $p = 0.32$, $r^2 = 0.02$) nor when we considered the two group separately (SC: $r = -0.18$, $p = 0.44$, $r^2 = 0.03$; EB: $r = -0.30$, $p = 0.2$, $r^2 = 0.09$, Fig. S7). If something, the correlation between 6-fold symmetry estimates and accuracy in EB individuals showed the opposite (negative) trend, letting us conclude that the reduction of grid-like coding in EB’s EC could be unlikely explained by differences in accuracies between the groups.”

3. P. 6 – you show 6-fold signal in the left EC but not right EC (in the SC group), and then show no 4, 5, or 7 fold signal in left EC. For completeness, I think you should also show no 4, 5, or 7 fold in the right EC.

We are sorry that we provided incomplete information. The required information about the absence of significant BOLD modulation by alternative periodicities (i.e., 4-,5-,7-,8-Fold symmetries) in the right EC have been reported in the main text as follows (Page 8): “Moreover, we did not detect any significant modulation of the BOLD signal when alternative models were tested in the right EC Bonferroni-corrected across left, right and bilateral hemispheres; 19 SC; 4-fold: $t_{18} = -0.86$, $p = 0.19$, Cohen’s $d = 0.20$; 5-fold: $t_{18} = 1.23$, $p = 0.11$, Cohen’s $d = 0.28$; 7-fold: $t_{18} = 1.56$, $p = 0.06$, Cohen’s $d = 0.35$, and 8-fold: $t_{18} = -0.54$, $p = 0.29$, Cohen’s $d = 0.12$, all results one-tailed, $\alpha_{\text{Bonferroni}} = 0.016$.” These results are reported in the text but not inserted in Fig.3 to avoid overcrowding in the figure.

4. I understand why you used a non-parametric statistic to test for 6-fold signal in the EB group, however it does mean the stats you use differs across groups. For completeness, it would be good to show stats across both a parametric and non-parametric test for both groups, so the reader can compare the groups more directly

Taking into account the reviewer’s suggestions, we have now added the results of the Wilcoxon signed-rank test to the main manuscript as follows (page 8): “(For completeness, if we apply the Wilcoxon test to SC data mentioned in the previous paragraph, results did not change: Bonferroni-corrected for multiple comparisons across left, right, and bilateral hemispheres; Left EC: $V = 154$, $z = -2.35$, $p = 0.0078$, $r = 0.54$; Right EC: $V = 95$, $z = 0$, $p = 0.5$, $r = 0$, all results one-tailed, $\alpha_{\text{Bonferroni}} = 0.016$.”

5. You show the 4-fold signal in EB is greater than in SC, however there isn’t a difference in the 6-fold signal in SC relative to EB. For completeness, you could present the interaction between the two (i.e., (4-fold EB – SC) – (6-fold EB – SC)).

We thank the reviewer for the thoughtful comment. It is certainly possible to test for a 2-way interaction. However, it might not be optimal. First, the study was not designed with a 2-by-2 design, and the comparison between the 6-fold and the 4-fold conditions would be essentially post hoc. Second, the 6-fold signal in the sighted is unilateral (i.e., the left entorhinal cortex), whereas the 4-fold signal in the blind is bilateral. This is not a problem with the current design and analysis, but it becomes a problem when looking for an interaction since there are three possible ways to proceed (note that since the data in 2 out of 3 cases were not normally distributed, we could not run an ANOVA, and the Wilcoxon test does not allow for interaction analysis. Thus, we subtracted the value of 4-fold from 6-fold for each participant in each group and then tested for group difference):

- (I) First, we've calculated the difference between 4- and 6-fold symmetries estimates in the bilateral EC for both groups and computed a Wilcoxon test on the obtained values since they were not normally distributed (Shapiro-Wilk normality test; $W = 0.78$, $p = 4.737e-6$). Results of the Wilcoxon test showed a significant difference between the two groups ($W = 101$, $z = -2.32$, $p = 0.01$, two-tailed).
- (II) Second, considering that we have a significant 6-fold symmetry in the left EC of SC participants and a significant 4-fold symmetry in the bilateral EC of EB participants, we have performed a similar analysis as above by computing the difference between 4- and 6-fold symmetry in the left EC for SC and bilateral EC for blind. Once again, we performed a Wilcoxon test as the obtained values were not uniformly distributed ($W = 0.8$, $p = 1.007e-5$). Similarly to the results reported above, the Wilcoxon test detected a significant difference between the two groups (Wilcoxon test: $W = 265$; $z = -2.49$, $p = 0.013$, two-tailed).
- (III) Lastly, we've also performed that same analysis using the 4- and 6-fold symmetry estimates obtained in the left EC of both groups. In this case, the difference between 4-fold and 6-fold symmetry was normally distributed ($W = 0.95$, $p = 0.11$). Therefore, we computed a two-sample t-test, which revealed a marginally significant difference between the two groups (two-sample t-test: $t_{36} = -1.81$, $p = 0.078$; Wilcoxon test: $W = 245$, $z = -1.87$, $p = 0.06$; two-tailed), though this result was significant one-tailed (two-sample t-test: $p = 0.039$; Wilcoxon test : $p = 0.03$,allowed given the post hoc nature of this comparison).

Given the post hoc nature of these analyses, we would not be keen to report them in the text. However, if we do, we think that the best alternative would be the simple effect analysis computed on the bilateral EC estimates for both groups, as it is the one carrying less *a priori* assumption.

We hope that we addressed the reviewer's comment and that they will understand our hesitation in reporting this analysis in the text straight away. But we remain open to suggestions.

6. P. 9 – you present evidence for a correlation between the 4-fold signal and activity in IPC in the EB but not SC groups. It would strengthen your findings to show that there is a significant difference between the slopes of these two correlations.

We thank the reviewer for this suggestion, which we've addressed by computing a multiple linear regression. We, therefore, model the 4-fold symmetry estimate in the EC as a function of the IPC activation, Group, and their interaction. As shown in the table below, we found a significant effect of IPC activity (i.e., the IPC activity predicts the magnitude of the 4-Fold symmetry; $p = 0.005$, two-tailed) and a significant effect of Group (i.e., There is a significant difference in the magnitude of the 4-fold symmetry between the two groups; $p = 0.001$, two-tailed). Moreover, there was a significant interaction effect between IPC Activity and Group, suggesting that the correlation between the 4-fold symmetry and IPC activity was higher in the EB group compared to the SC group ($p = 0.02$, two-tailed).

4-fold Symmetry Score Predicted by IPC Activity

Coefficients	Estimate	Std. Error	T-value	Pr (> t)
Intercept	0.311	0.085	3.656	0.0008***
IPC Activity	0.153	0.052	2.95	0.005**
Group	- 0.41	0.12	- 3.4	0.0017**
IPC Activity : Group	- 0.18	0.07	- 2.33	0.02*

$R^2 = 0.38$; *Adjusted R*² = 0.32; $F(3,34) = 6.95$; $p = 0.0009$.

These results have been integrated with the main manuscript as follows (page 10):

“Furthermore, we performed a linear regression analysis to investigate whether this effect was stronger in EB compared to SC. Indeed, we detected a significant interaction between IPC activity and groups (t-value = -2.33, $p = 0.02$, $r^2 = 0.38$ two-tailed, see Supplementary Table 5) strengthening the hypothesis that the use of an egocentric perspective to navigate the environment influenced the neural geometry of entorhinal cognitive maps in EB”

Moreover, although not requested by the reviewer, for completeness, we performed two additional multiple linear regression models to investigate whether the accuracy (arcsin transformed) in the Clock Navigation task and in the Path Integration task would have been significantly better predicted by IPC activity in EB rather than SC.

First, the results from the linear model, which aimed to investigate the relation between accuracy in the Clock-Navigation task and the IPC activity, revealed a significant effect of IPC Activity (i.e., IPC activity predicted the accuracy in the clock-navigation task; $p = 0.048$, two-tailed), a significant effect of Group (i.e., There was a significant difference in the performance during the task between the two groups; $p = 0.03$, two-tailed) and a marginally significant interaction effect between IPC Activity and Group (i.e., IPC activity

seems to predict differently the accuracy during the task across the two groups; $p = 0.079$, two-tailed).

Coefficients	Estimate	Std. Error	T-value	Pr (> t)
Intercept	1.30	0.02	62.55	<2e-16***
IPC Activity	0.03	0.01	2.04	0.048*
Group	0.07	0.02	2.25	0.03*
IPC Activity : Group	- 0.03	0.02	- 1.8	0.079.

$R^2 = 0.21$; Adjusted $R^2 = 0.14$; $F(3,34) = 3$; $p = 0.039$

Second, the linear model conducted aiming to investigate the relation between Path Integration accuracy and IPC activation revealed a significant effect of IPC Activity (i.e., the IPC Activity predicted the accuracy during path integration; $p = 0.008$, two-tailed), a significant effect of Group (There was a significant difference in the path integration performance across the two groups; $p = 0.02$, two-tailed) and a marginally significant interaction effect between IPC Activity and Group (i.e., IPC activity seems to correlate differently with path integration abilities across the two groups; $p = 0.059$, two-tailed).

Coefficients	Estimate	Std. Error	T-value	Pr (> t)
Intercept	0.27	0.007	38.16	<2e-16***
IPC Activity	0.01	0.004	2.81	0.008**
Group	0.02	0.01	2.44	0.02*
IPC Activity : Group	- 0.01	0.006	- 1.95	0.059.

$R^2 = 0.29$; Adjusted $R^2 = 0.23$; $F(3,34) = 4.52$; $p = 0.009$.

In summary, the two additional linear models, computed to investigate whether IPC activity could predict the accuracy in the Clock Navigation experiment and in the Path Integration experiment better in EB than in SC, showed a marginally significant interaction between IPC Activity and Group. We've integrated these results in the revised manuscript as follows:

a. Linear model on accuracy in the Clock Navigation experiment (page 10): **“However, in this case the interaction between IPC activity and Group was only marginal (t-value = - 1.8, $p = 0.08$, $r^2 = 0.21$, two-tailed).”**

b. Linear model on accuracy during the Path Integration experiment (page 14): **“Comparing the two groups with linear regression analysis we found a marginally significant IPC-activity by Group interaction (17 SC & 19 EB, t-value = -1.95, $p = 0.059$, $r^2 = 0.29$ two-tailed, Fig. 4E).”**

Given that, in the case of the correlation with accuracy, the difference between sighted and blind was only marginal, we've decided to avoid any strong interpretation of these

results. Thus, we deleted the heading: “Parietal cortex activity predicted real-world navigation performance in early blind”, and we added a cautionary note on the interpretation of such correlations (p. 14):

“Although correlations performed with a limited number of subjects should be taken with caution, these results may be considered as preliminary evidence that an increased activity in inferior parietal cortex, in the EB, is associated with increased navigation performances both in imagined and real-life navigation.”

7. P. 13 – “which would not have demanded a strong reliance on episodic memory to retrieve its spatial configuration”. I find this argument slightly odd, given the strong emphasis of the paper on cognitive maps in the hippocampal-entorhinal system, to then dismiss a lack of hippocampal activation in terms of episodic memory. I sympathise with the argument, but it might need to be articulated a bit more carefully.

We thank the reviewer for the comment. In the revised manuscript, we articulated our argument as follows (page15):

“Indeed, several studies have shown that hippocampal activity decreases during the exploration or recall of a familiar environment compared to novel ones (Kaplan et al., 2014; Cheng et al., 2008).”

Reviewer #2 (Remarks to the Author):

Sigismondi and colleagues investigated entorhinal grid-like codes in early blind people. Given that the hippocampal-entorhinal system is linked to the construction of allocentric spatial-cognitive maps, primarily driven by visual input in humans, the authors asked whether early blind individuals would show altered grid-like codes during (imagined) navigation through a clock space compared to sighted controls.

They found that although both early blind and sighted individuals engaged brain regions typically involved in spatial navigation (Human Navigation Network) to a similar extent, grid-like codes were not significantly increased in early blind individuals. With additional analyses, they showed that inferior parietal activation appears increased during navigation in blind individuals and that this effect is further linked to increased clock navigation performance, path integration, and a 4-fold (rather than a 6-fold) rotational symmetry of the entorhinal BOLD signal (which they discuss as potential evidence for egocentric rather than allocentric navigation strategies in early blindness).

The results are interesting and would certainly have an impact on the field, the paper is well-written. My main comments pertain to the very small (and not significant) difference in (6-fold) grid-like codes between the groups – making it difficult to clearly decide whether grid-like codes are actually altered in early blind compared to sighted individuals

Main comments:

Clock navigation experiment: Did the participants receive any instruction on how large they should imagine the clock-like space or with what walking speed they should imagine to navigate? For instance, imagining a smaller space would lead to shorter path distances (which could be coupled to lower grid signals). Did participants have to indicate via button press when they completed the trajectory? This was not clear to me from the task description.

We thank the reviewer for the comment, and we apologize for the unclear description of the task. Participants did not receive any explicit information about the size of the clock or the velocity with which they should have walked within it, as it would have been difficult to instruct participants to adopt a predefined speed (e.g., 1 step per second). Instead, we have made participants aware that they had about 4 seconds to perform the imagined navigation. Although this instruction does not ensure that participants were “walking” at the same speed for the same amount of time, it provides at least a predefined time window within which the trajectory should be performed. Participants were not required to indicate the completion of the trajectory via button press. This information has been added in the method section (page 19).

“Participants did not receive any instruction concerning the size of the clock or the speed with which they should have had navigated through it, however, they were aware that the imagination period could last a maximum of four seconds.”

Concerning the size of the imagined clock, although it could vary across participants, we cannot think of a clear reason why it should be different across groups, especially given the identical instructions provided. Moreover, grid cells are organized in modules with different spatial resolutions, which allows for tiling and navigating spaces with different sizes (Stensola et al., 2012), and grid-like coding also emerges during conceptual navigation (Constantinescu et al., 2016; Park et al., 2021) in which the size of the space is, by definition, not definable and, arguably, may be different across subjects. Finally, the main result of the paper is not an absence of grid-like activity in the blind (that might have triggered this doubt in the reviewer’s mind) but a deformation of the geometry of cognitive maps, showing a double dissociation across groups (6-fold vs 4-fold). Four-fold symmetry, in humans (He et al., 2000; Wagner et al., 2023) and animals (Derdikman et

al., 2009), in previous studies, emerges in different conditions (e.g., with or without barriers) in environments of the same exact size, and what seems to be in common across all these studies is a higher reliance on an egocentric referential frame (see the *discussion* section), which we consider the best candidate to explain the 4-fold symmetry (as well as the connected results: anchoring of grid orientation, involvement of parietal cortex).

However, in light of the reviewer's comment, we decided to highlight the fact that we could not completely control participants' navigation behavior as a limitation of the study in the discussion session as follows (page 18):

“It remains possible that our results could, in part, be related to the specifics of the imagined clock environment, and further studies using different types of environments, as well as auditory or tactile navigation (Ottink et al., 2021; in which navigation behavior can be better controlled) are needed to fully understand the differences between sighted and early blind during spatial navigation.”

Close navigation experiment, accuracy: In fig 3c it seems that there is an outlier in the sighted control group (value between 0.8-0.85%). If so, this would likely contribute to the non-significant difference between SC and EB (in other words, it could be problematic if EB performance is significantly lower than SC after outlier exclusion, as the performance difference could explain the non-significant grid-like codes).

We thank the reviewer for the thoughtful comment. As reported above (see reviewer 1 point 2), EB participants' performance seems to be slightly lower than the performance of SC participants. Indeed, removing the outlier in the SC groups resulted in a significant difference between the performance of the two groups (two-sample t-test; $t_{35} = 2.87$, $p = 0.007$). Similarly, the Bayesian test conducted on the full sample (19 SC & 19 EB) reported very weak evidence in favor of a difference in the accuracy of the two groups ($BF_{10} = 1.237$; error % = 0.006, Kass et al., 1995) but substantial evidence in favor of it, when we removed the outlier in the SC group (18 SC & 19 EB, $BF_{10} = 6.665$; error % = 2.6×10^{-6} , Kass et al., 1995). However, as already highlighted above in response to a similar concern raised by Reviewer #1, the reduced grid-like coding in EB participants was hardly attributable to the difference in performance between the two groups. First, the overall accuracy in both SC and EB individuals was high (about 95% and 92%, respectively), denoting the ability of both groups to solve the task. Secondly, correlation analysis revealed that there was no correlation between the accuracy score and magnitude of the 6-fold symmetry in either of the groups (SC: $r = -0.18$, $p = 0.44$, $r^2 = 0.03$; EB: $r = -0.30$, $p = 0.2$, $r^2 = 0.09$, see reviewer 1 point 2). As shown in Fig. S7, there was actually a tendency for a negative correlation between 6-fold symmetry estimates and accuracy in EB individuals, i.e., participants with the lower 6-fold symmetry estimate

were also those with higher accuracy. In sum, these additional analyses strongly speak for the absence of a link (correlation) between the accuracy in the clock navigation task and the lack/presence of 6-fold symmetry in both groups. We've integrated the above reported results for accuracy in the Clock-Navigation experiment and correlation between accuracy and grid-like signal in the main manuscript under the **“Blind and sighted successfully navigated the clock space”** (page 3):

“No difference in accuracy was detected between groups and tasks in the Nav-Math experiment (Repeated-measure ANOVA; main effect of group: $F_{(1,36)} = 0.86$, $p = 0.36$, $\eta^2 = 0.018$; main effect of task: $F_{(1,36)} = 0.85$, $p = 0.36$, $\eta^2 = 0.005$; Group \times Task interaction: $F_{(1,36)} = 1.42$, $p = 0.24$, $\eta^2 = 0.008$, two-tailed; Fig. 1D). Reaction times (RTs) followed a similar pattern with no differences detected in RTs between groups and tasks in the Nav-Math experiment (main effect of group $\chi^2_{(1)} = 1.53$, $p = 0.21$; main effect of task: $\chi^2_{(1)} = 3.59$, $p = 0.06$; Group \times Task interaction: $\chi^2_{(1)} = 0.24$, $p = 0.62$, two-tailed, Fig. 1F). Collectively, these results suggest that both sighted and blind were able to perform the tasks and to perform comparably.

Similarly, in the Clock-Navigation experiment, we did not detect differences in accuracy (two-sample t-test; $t_{(36)} = 2.0$, $p = 0.052$, Cohen's $d = 0.62$, two-tailed, Fig. 1C) and RTs (Linear Mixed-Effect Model; $\chi^2_{(1)} = 2.45$, $p = 0.12$, two-tailed, Fig. 1E) between Groups. However, outlier analysis revealed the presence of one outlier in accuracy in the SC group. Thus, we've performed the analyses excluding the outlier participants. Results on the RTs remain invariant with no differences between groups (Linear Mixed-Effect Model; 18 SC & 19 EB, $\chi^2_{(1)} = 2.28$, $p = 0.13$, two-tailed). On the other hand, analyses on accuracy detected a significant difference between the two groups (18 SC & 19 EB, $t_{(35)} = 2.87$, $p = 0.007$, Cohen's $d = 0.94$, two-tailed). Despite this slight difference in navigation accuracy, it is important to consider that the average accuracy was very high in both groups (19 SC, accuracy % = 95; 19 EB, accuracy % = 92), therefore we could conclude that both sighted and blind participants were able to successfully navigate the clock environment during the Clock-Navigation experiment.”

And the **“Six-fold grid-like coding did not emerge in early blind”** (page 8-9):

“Accounting for the possible difference in clock-navigation accuracy between the two groups (see above), we'd furthermore investigated whether the reduced grid-like activity in EB participants' EC was attributable to this putative difference. We did not detect any significant correlation between accuracy during the Clock-Navigation experiment and the magnitude of 6-Fold symmetry estimates not when combining the two groups (19 SC & 19 EB, $r = -0.16$, $p = 0.32$, $r^2 = 0.02$) nor when we considered the two group separately (SC: $r = -0.18$, $p = 0.44$, $r^2 = 0.03$; EB: $r = -0.30$, $p = 0.2$, $r^2 = 0.09$, Fig. S7). If something, the correlation between 6-fold symmetry estimates and accuracy in EB individuals

showed the opposite (negative) trend, letting us conclude that the reduction of grid-like coding in EB's EC could be unlikely explained by differences in accuracies between the groups."

It is difficult to interpret the meaning of significantly increased grid-like codes in SC but not EB without a significant group difference (i.e., the fact that SC is significant but not EB does not necessarily mean that SC > EB, or that grid-like codes in EB are not present). It would be helpful if the authors could confirm the null result using Bayesian statistics to ameliorate some of the concerns related to the very similar grid-like values in SC and EB. This point makes it difficult to state that grid-like codes in EB are truly altered.

We thank the reviewer for the comment, and as suggested, we've performed a Bayesian test in the early blind, which reported moderate evidence in favor of the null hypothesis (absence of a six-fold effect in EB in the Left EC $BF_{10} = 0.23$; error % = 0.016; Right EC: $BF_{10} = 0.24$, error % = 0.016; Bilateral EC: $BF_{10} = 0.27$, error % = 0.017). The results had been added to the ' **Six-fold grid-like coding did not emerge in early blind**' paragraph (pages 8):

"However, the lack of hexadirectional coding in EB was strengthened by Bayesian analysis showing no evidence of six-fold symmetry in this group (Left EC $BF_{10} = 0.23$; error % = 0.016; Right EC: $BF_{10} = 0.24$, error % = 0.016; Bilateral EC: $BF_{10} = 0.27$, error % = 0.017)."

To conclude, we would like to highlight that the statement that the grid-like code is altered refers to the fact that we found a 90° grid (4-fold) in the blind and not in the sighted, with a significant interaction across groups (19 SC, 19 EB; Two-sample t-test; $t_{36} = -3.18, p =$

0.003, Cohen's $d = 1.03$, two-tailed, $\alpha = 0.05$; Fig. 3G). In other words, the absence of a hexagonal symmetry is not equivalent to the absence of a grid-like code, as rightly pointed out by the reviewer. Instead, the alteration of the neural geometry of cognitive maps can lead to a grid code with a different periodicity (e.g., 90°) and not only to a loss of periodicity tout court. A well-known example is the alteration of the spatial periodicity of grid-cell firing fields in hairpin mazes, both in rodents and humans (He et al., 2019; Derdikman et al., 2009). In this case, as in the current paper, grid cells' geometry was altered (from 60° to 90°) but still consistently periodic. In sum, this is why we refer to the grid code as "altered" instead of "absent". We made it more clear in the manuscript (page 9):

"In sum, although a significant 6-fold symmetry emerged in SC (as predicted) and did not in EB, given the lack of a significant between-group difference, we can't conclude that the typical hexadirectional grid code in EB is different from the one in SC. Nevertheless, it might be that the reduced grid-like coding in EB participants' EC is related to an alteration of the typical hexadirectional geometry that could give rise to an alternative neural geometry. We've tested this hypothesis in the following paragraph."

Minor comments:

Abstract, line 12, there are two commas " , , "

We've fixed the issue.

Abstract, line 15, "Human Navigation Network", it might not be clear to every reader what this entails. I would suggest writing out (at least the core regions).

We thank the reviewer for having pointed out this issue. We've addressed it by mentioning in the abstract the core regions that belong to the Human Navigation Network as follows (page 1): "During imagined navigation, the Human Navigation Network, constituted by frontal, medial temporal, and parietal cortices, was reliably activated in both groups, showing resilience to visual deprivation."

Introduction, line 61, whether fMRI-based grid-like codes are "a reliable proxy for the activity of grid cells" is debated in the field and I would suggest to phrase this more carefully

We thank the reviewer for the comment, and we addressed it by toning down the sentence in the manuscript as follows (page 2):

“The clock space was sampled at a granularity of 15° to provide a sufficient angle resolution to detect the hexadirectional signal (6-fold symmetry, Fig. 1A), which some studies³⁶⁻⁴⁰, have indicated as a proxy for the activity of grid cells in fMRI both during visual³⁶⁻³⁸ and imagined navigation^{39,40}.”

Results, please report sample sizes and effect sizes for each analysis step / result

We thank the reviewer for the comment. Accounting for the statistical test used for each analysis, we have reported Cohens' d for the t-test, η^2 for the repeated measure ANOVA, r for Wilcoxon tests, and r^2 for correlations. We've highlighted all the changes throughout the manuscript in red. However, in order not to make the result section redundant and overcrowded, we have specified the sample size at the end of the manuscript (“**Statistical analyses**” page 27) and reported it in the main text just when it was different (for example the case of Path Integration experiment analyses). Moreover, we have decided not to specify the effect size for the linear mixed models analyses as it is currently debated, in the field, which index would be the most appropriate. We hope, however, that this will be enough to address reviewer concerns.

I was not able to find the description in the text belonging to Fig. 3H. I think there is something missing?

We thank the reviewer for the comment, and we apologize for the lack of clarity concerning the absence of a description in the main text of Fig. 3H. We did not cite Fig. 3H in the main text as the image was integrated just for displaying purposes with the aim to show the readers the approximate location of the cluster found in the EC of SC and EB participants for the fold of interest, 4-fold and 6-fold respectively (similarly to what Doeller and colleagues did in their 2010 paper). In the revised manuscript we now refer to the image when we describe ROI results line 199-201 and 249 for SC and EB respectively. Moreover, a brief description of how the images have been obtained can be found in the method section (lines 684 to 686).

Reviewer #3 (Remarks to the Author):

The manuscript describes extensive work on spatial cognitive maps in early blind versus sighted controls. Two types of analysis on fMRI data are combined with behavioural data. The experiments are carefully performed and well executed, and processing of data is fine. The experiments are well described and there is enough detail in the methods to understand how the experiments were performed. The main concern is statistics (see main comment). The main conclusion is that early blind show a 90 degrees rotational

symmetry (4-fold) and controls a 60 degrees rotational symmetry. The results for the early blind are significant for both hemispheres, while the results for the controls are only significant for the left hemisphere

Main point: Bonferroni corrections are applied, but inconsistently, and in an unclear way. In principle, the comparison that is made in the experiment for the main conclusion is for 3 types of analysis (bilateral, left hemisphere and right hemisphere) and for 5 models (4, 5, 6, 7, and 8 folds). This would in principle mean a bonferroni correction for $5 \times 3 = 15$ comparisons), but if this would be applied no statistical differences can be reported for any of the results. In the sighted control condition the bonferroni correction is applied only for hemisphere conditions (3). For the early blind it is applied for four tested periodicities. Why this difference for SC vs EB? Why in the case of EB only four, since there were 5 periodicity models tested? Also the correction is for four cases according to the results section and the p-value is 0.025, why is that called a significant effect? In principle, the fact that results are insignificant is not a problem when there is a very clear hypothesis to start with. For the 6-fold model in sighted controls this might be the case, also because this has been reported before in previous studies. However, the strong conclusion about the 4-fold representation in early blind looks more like a post-hoc finding, without properly correcting for the number of comparisons. By taking this into account the manuscript has to be drastically rewritten.

We thank the reviewer for the comment. First of all, the p-values reported in the original manuscript are Bonferroni-corrected ones, not raw p-values. This seems to have generated some confusion, and indeed, we realized that it might not be the best way to report the results. Throughout the paper, we now report the raw p-values (one-tailed, unless differently specified) and declare the Bonferroni-corrected alpha threshold. This should address the following part of the reviewer's comment.

"Also the correction is for four cases according to the results section and the p-value is 0.025, why is that called a significant effect?"

To further clarify, "p = 0.025" in that sentence refers to the already Bonferroni-corrected p-value (the raw p-value is 0.006). Thus, the p-value survived correction for 4 multiple comparisons.

Concerning the correction for multiple comparisons, we tried to be as conservative as possible using Bonferroni correction instead of less conservative methods and correcting for all the possible comparisons in each analysis. However, we agree with the reviewer that we can improve this procedure. Concerning the six-fold effect in the sighted entorhinal cortex, we now correct for three multiple comparisons (bilateral, left, and right

ROI) instead of two, as suggested by the reviewer. Results do not change (6-fold: 19 SC, $t_{18} = 2.71$, $p = 0.007$, Cohen's $d = 0.62$, one-tailed, $\alpha_{\text{Bonferroni}} = 0.016$). As correctly pointed out by the reviewer, six-fold symmetry was predicted based on previous results in sighted participants, and none of the control folds was considered a plausible alternative but simply a control (see References Doeller et al., 2010; Horner et al., 2016; Stangl et al., 2018; Nau et al., 2018); thus no further correction was applied.

However, for the 4-fold effect in the blind, the prediction was not as straightforward. Although 4-fold symmetry has been reported in previous studies (He et al., 2019; Wagner et al., 2023), and that is not the case for other symmetries (e.g., 5-fold, 7-fold, 8-fold), we did not necessarily predict it, and we deemed necessary to correct for multiple comparisons with the other folds. In the original manuscript, we left out the 6-fold periodicity from this count, correcting for four multiple comparisons (4-, 5-, 7-, 8-fold). However, we agree with the reviewer that it is better to correct for 5 comparisons (i.e., including 6-fold). We did it in the new version of the manuscript, and the results did not change (4-fold: 19 EB, $t_{18} = 2.77$, $p = 0.006$, Cohen's $d = 0.64$, $\alpha_{\text{Bonferroni}} = 0.01$).

As the reviewer pointed out, correcting for 15 multiple comparisons in the case of EB subjects will weaken the 4-fold effect, which will become, at best, marginal ($\alpha_{\text{Bonferroni}} = 0.0033$, $p=0.006$, one-tailed). However, we do not think such a correction is necessary since we did not predict a lateralization of the 4-fold effect, and indeed we found the effect in the bilateral ROI with no difference across hemispheres. In addition to that, further analysis shows that the 4-fold effect is accompanied by spatial anchoring (Fig. 3I) and is predicted by parietal activity (Fig. 3L), further strengthening the reliability of this finding.

We hope that our clarifications and the modifications of the result and the method sections ("**Statistical analyses**" paragraph page 27) following the Reviewer's suggestions address all the expressed concerns.

Minor point: Second sentence in the abstract has two comma's.

We've deleted one comma.

Further changes:

- Added details to the Figure S1

References:

1. Bellmund, J. L., Deuker, L., Navarro Schröder, T., & Doeller, C. F. (2016). Grid-cell representations in mental simulation. *Elife*, 5, e17089.

2. Cheng, S., & Frank, L. M. (2008). New experiences enhance coordinated neural activity in the hippocampus. *Neuron*, 57(2), 303-313.
3. Kanjlia, S., Lane, C., Feigenson, L., & Bedny, M. (2015). Visual cortex of congenitally blind individuals responds to symbolic number. *Journal of Vision*, 15(12), 194-194.
4. Derdikman, D., Whitlock, J. R., Tsao, A., Fyhn, M., Hafting, T., Moser, M. B., & Moser, E. I. (2009). Fragmentation of grid cell maps in a multicompartiment environment. *Nature neuroscience*, 12(10), 1325-1332.
5. Doeller, C. F., Barry, C., & Burgess, N. (2010). Evidence for grid cells in a human memory network. *Nature*, 463(7281), 657-661.
6. Gagnon, L., Schneider, F. C., Siebner, H. R., Paulson, O. B., Kupers, R., & Ptito, M. (2012). Activation of the hippocampal complex during tactile maze solving in congenitally blind subjects. *Neuropsychologia*, 50(7), 1663-1671.
7. Kass, R. E., & Kass, A. E. (1995). Bayes Factors. *Journal of the American Statistical Association*, 90(430), 773–795. <https://doi.org/10.2307/2291091>
8. Kaplan, R., Horner, A. J., Bandettini, P. A., Doeller, C. F., & Burgess, N. (2014). Human hippocampal processing of environmental novelty during spatial navigation. *Hippocampus*, 24(7), 740-750.
9. Kupers R, Chebat DR, Madsen KH, Paulson OB, Ptito M. Neural correlates of virtual route recognition in congenital blindness. *Proceedings of the National Academy of Sciences of the United States of America*. 2010;107(28):12716-12721. doi:10.1073/pnas.1006199107
10. Stensola, H., Stensola, T., Solstad, T., Frøland, K., Moser, M. B., & Moser, E. I. (2012). The entorhinal grid map is discretized. *Nature*, 492(7427), 72-78.
11. Nau, M., Navarro Schröder, T., Bellmund, J. L., & Doeller, C. F. (2018). Hexadirectional coding of visual space in human entorhinal cortex. *Nature neuroscience*, 21(2), 188-190.

Reviewer #1 (Remarks to the Author):

The authors have responded thoroughly to my previous comments and I think the manuscript has been strengthened as a result. This really is an interesting study that must have been technically demanding to carry out, so I commend the authors for their work. I have no further comments.

Reviewer #2 (Remarks to the Author):

The authors addressed all my concerns and I have no further comments.

Reviewer #3 (Remarks to the Author):

The authors addressed the concerns and all the changes made in the manuscript now have strengthened the paper. I have no further comments.